# LNK suppresses interferon signaling in melanoma

Ling-Wen Ding[1], Qiao-Yang Sun [1], Jarem J. Edwards[2,3], Lucia Torres Fernández[1], Xue-Bin Ran[1], Si-Qin Zhou[1], Richard A. Scolyer [2,3,4], James S. Wilmott[2,3], John F. Thompson[2,3,4], Ngan Doan[5], Jonathan W. Said [5], Nachiyappan Venkatachalam [1], Jin-Fen Xiao[1], Xin-Yi Loh[1], Maren Pein[1], Liang Xu[1], David W. Mullins [6], Henry Yang[1], De-Chen Lin[7] & H. Phillip Koeffler [1,7]

LNK (SH2B3) is a key negative regulator of JAK-STAT signaling which has been extensively studied in malignant hematopoietic diseases. We found that LNK is significantly elevated in cutaneous melanoma; this elevation is correlated with hyperactive signaling of the RAS-RAF-MEK pathway. Elevated LNK enhances cell growth and survival in adverse conditions. Forced expression of LNK inhibits signaling by interferon-STAT1 and suppresses interferon (IFN) induced cell cycle arrest and cell apoptosis. In contrast, silencing LNK expression by either shRNA or CRISPR-Cas9 potentiates the killing effect of IFN. The IFN-LNK signaling is tightly regulated by a negative feedback mechanism; melanoma cells exposed to IFN upregulate expression of LNK to prevent overactivation of this signaling pathway. Our study reveals an unappreciated function of LNK in melanoma and highlights the critical role of the IFN-STAT1-LNK signaling axis in this potentially devastating disease. LNK may be further explored as a potential therapeutic target for melanoma immunotherapy.

[1] Cancer Science Institute of Singapore, National University of Singapore, Singapore 117599, Singapore. [2] Melanoma Institute Australia, The University of Sydney, Sydney, NSW 2065, Australia. [3] Sydney Medical School, The University of Sydney, Sydney, NSW 2006, Australia. [4] Royal Prince Alfred Hospital, Sydney, Sydney, NSW 2050, Australia. [5] Santa Monica-University of California, Los Angeles Medical Center, Los Angeles, CA 90095, USA. [6] Departments of Medical Education and Microbiology/Immunology, Geisel School of Medicine at Dartmouth, Dartmouth, MA 03755, USA. [7] Division of Hematology/Oncology, Cedars-Sinai Medical Center, UCLA School of Medicine, Los Angeles, CA 90048, USA. Correspondence and requests for materials should be addressed to Q.-Y.S. (email: qiaoyangsun@gmail.com)

Advanced stage melanoma is one of the most devastating cancers causing nearly 50,000 annual deaths worldwide[1]. The majority of cutaneous melanomas are driven by oncogenic gain of function mutant BRAF (V600E, ~40–50%)[2,3], followed by mutant NRAS, RAS inhibitor NF1 (loss of function) and occasionally mutant c-Kit. Melanoma cells generally respond poorly to traditional chemotherapy and radiotherapy, until recently rendering a lack of efficient therapeutic approaches for most of the late stage patients. Development of BRAF inhibitors vemurafenib (PLX4032)[4–6] and dabrafenib and immune checkpoint blockade antibodies which target either cytotoxic T lymphocyte–associated antigen 4 (CTLA-4) or programmed cell death protein 1 (PD-1)[7,8] have dramatically altered the therapeutic landscape of melanoma in the past few years[9,10]. These approaches, particularly antibodies against immune checkpoint blockade, show strikingly durable responses resulting in significant improvement of overall survival of a subset (30–40%) of patients[8]. Despite these remarkable achievements, resistance to therapy and melanoma recurrence often occurs[11]. Specifically, about 60–80% of the melanoma patients do not response to initial PD-1/CTLA-4 antibody therapy[12,13] (primary resistance), and 20–30% of the initial responders develop resistance to treatment, with progressive disease[14]. Hence, a better understanding is clearly needed in regards to the signaling pathways governing the survival of melanoma cells in the setting of immunotherapy.

Recently, two studies noted the crucial role of the interferon (IFN) pathway in melanoma patients who become resistant to either PD-1 or CTLA-4 antibody therapy[15,16]. Loss of function mutations of JAK1/2 were found in patients with recurrent melanoma who initially responded to PD-1 antibody, or in the primary nonresponder (patients who never responded to the therapy)[17]. Functional studies undertaken by the same researchers suggested that loss of JAK2 function in melanoma cells could impairs IFN signaling, causing resistance to T cell-mediated cytotoxicity and thereby leading to recurrence of the melanoma[15,17]. Similar conclusions have been reached using in vivo CRISPR-Cas9 library screening studies showing that sgRNAs targeting Jak1/Jak2 and Stat1 as well as interferon receptors were significantly enriched in murine melanoma B16 cells placed in immune-competent, syngeneic C57B/L mice, compared to the same cells (with the same sgRNA library pool) grafted in immunodeficient mice[18,19]. These studies underscore the crucial role of IFN-JAK/STAT1 signaling in the immune escape of melanoma cells, consistent with dysregulation of the JAK-STAT signaling pathway facilitating progression of melanoma. Loss of the JAK-STAT signaling provides a selective growth/survival advantage for melanoma cells to thwart immune surveillance allowing negative modulators of this signaling to be explored as a potential therapeutic target.

LNK (SH2B3) is a key negative regulator of JAK-STAT signaling, which has been extensively studied in malignant hematopoietic diseases[20–23]. As an adaptor protein, LNK recognizes and binds to activated, phosphorylated tyrosine proteins through its SH2 domain, resulting in the inhibition of these activated kinases. Within this context, LNK is a potent tumor suppressor in hematopoietic malignancies[22,24,25], as many hematopoietic cancers are mainly driven by gain of function receptor tyrosine kinase (RTK)[24]. For example, in myeloid proliferative disorder (MPD), a blood cancer which frequently (~90–95%) harbors the V617F gain of function mutant JAK2[26], LNK behaves as an antiproliferative effector by directly binding and suppressing the signaling of this mutant kinase[20,27]. Indeed, loss of function mutations of LNK occur in MPD patients (particularly those with wild-type JAK2)[27,28] and occasionally in Philadelphia chromosome (Ph)-like acute lymphoblastic leukemia (ALL)[29]. Most LNK studies have focused on its role in hematopoietic disease, often

using murine Lnk knockout models[23–25,30–32]. Although LNK is widely expressed in a variety of cancer cells (Fig. 1a), its function in solid tumors has not been fully explored. In this study, we find that LNK is highly expressed in melanoma, and aberrant elevation of LNK confers a selective survival advantage for melanoma cells against the anti-proliferative and pro-apoptotic effect of interferon. Our study identify LNK as a critical regulator of the IFN-STAT1 pathway; and aberrantly expressed LNK probably contributes to immune evasion and tumorigenesis of melanoma.

## Results

**LNK expression is significantly elevated in melanoma.** We analyzed *LNK* mRNA expression in the Cancer Cell Line Encyclopedia (CCLE) [http://www.broadinstitute.org/ccle], cBioPortal for Cancer Genomics [www.cbioportal.org/], Oncomine [https://www.oncomine.org] and NCBI GEO database [https://www.ncbi.nlm.nih.gov/geo/]. Since primary tumors often contain infiltrating T/B lymphocytes[33], which are known to express considerable level of LNK, we began our analysis with cancer cell line data because they lack infiltrating lymphocytes and stroma cells. Among the 881 different cancer cell lines in the CCLE database and 317 cancer cell lines in the CellLineNavigator database (E-MTAB-37, Transcriptomics for Cancer Cell Line Project), *LNK* mRNA is significantly upregulated in cutaneous (skin) melanomas (Fig. 1a, upper and middle panels). Consistently, among the >8000 RNA sequencing data from primay cancer samples in The Cancer Genome Atlas (TCGA), melanoma samples expressed the highest *LNK* mRNA (Fig. 1a, lower panel). Compared to either normal skin tissue (Supplementary Fig. 1, RNA sequencing of 473 sun-exposed normal skin samples and 387 non-sun-exposed normal skin samples, collected from the GTEX database [https://www.gtexportal.org]) or benign nevi, *LNK* mRNA was significantly upregulated in the melanoma samples, particularly in advanced stages of the disease (vertical growth phase, metastatic growth phase vs primary/in situ melanoma, Fig. 1b–d) and ranked as one of the top 1% overexpressed genes in melanoma (Riker Melanoma, Oncomine database, Fig. 1b). We performed western blot and immunohistochemistry (IHC) staining to confirm the elevated expression of LNK protein in melanoma. In melanoma tissue arrays (ME242a, obtained from Biomax Inc), all of the melanoma section cores ($n = 12$) were heavily stained with the LNK antibody (Fig. 1e), while considerable less staining was found in normal skin tissue. We further extended the IHC study to a large melanoma patient cohort (tissue arrays established at Melanoma Institute Australia, The University of Sydney, Australia) including 163 melanoma patient samples. Strong staining of LNK was observed (score = 3 or 2) in the majority of the melanoma samples (81%). In addition, when we separated the stage 3 patients (which contained the largest patient number with similar stage of disease for analysis) based on the LNK staining results, a trend toward inferior overall survival was found in patients with high LNK protein expression in their tumors (Supplementary Fig. 2).

**LNK expression is correlated with RAS-RAF signaling.** We sought to examine the underlying mechanism governing the aberrant elevation of LNK in melanoma. First, we analyzed the CCLE melanoma cell lines to assess whether any potential correlation occured between *LNK* and other oncogenic abnormalities. We separated the melanoma cell lines based on their driver mutation (BRAF V600E, NRAS Q61K/L, c-KIT or other), and found that *LNK* expression was significantly higher in cell lines that harbored BRAF and NRAS mutations (Fig. 1f), suggesting that hyperactivated RAS-RAF-MEK signaling may correlate with LNK expression in melanoma. To pursue this hypothesis, we

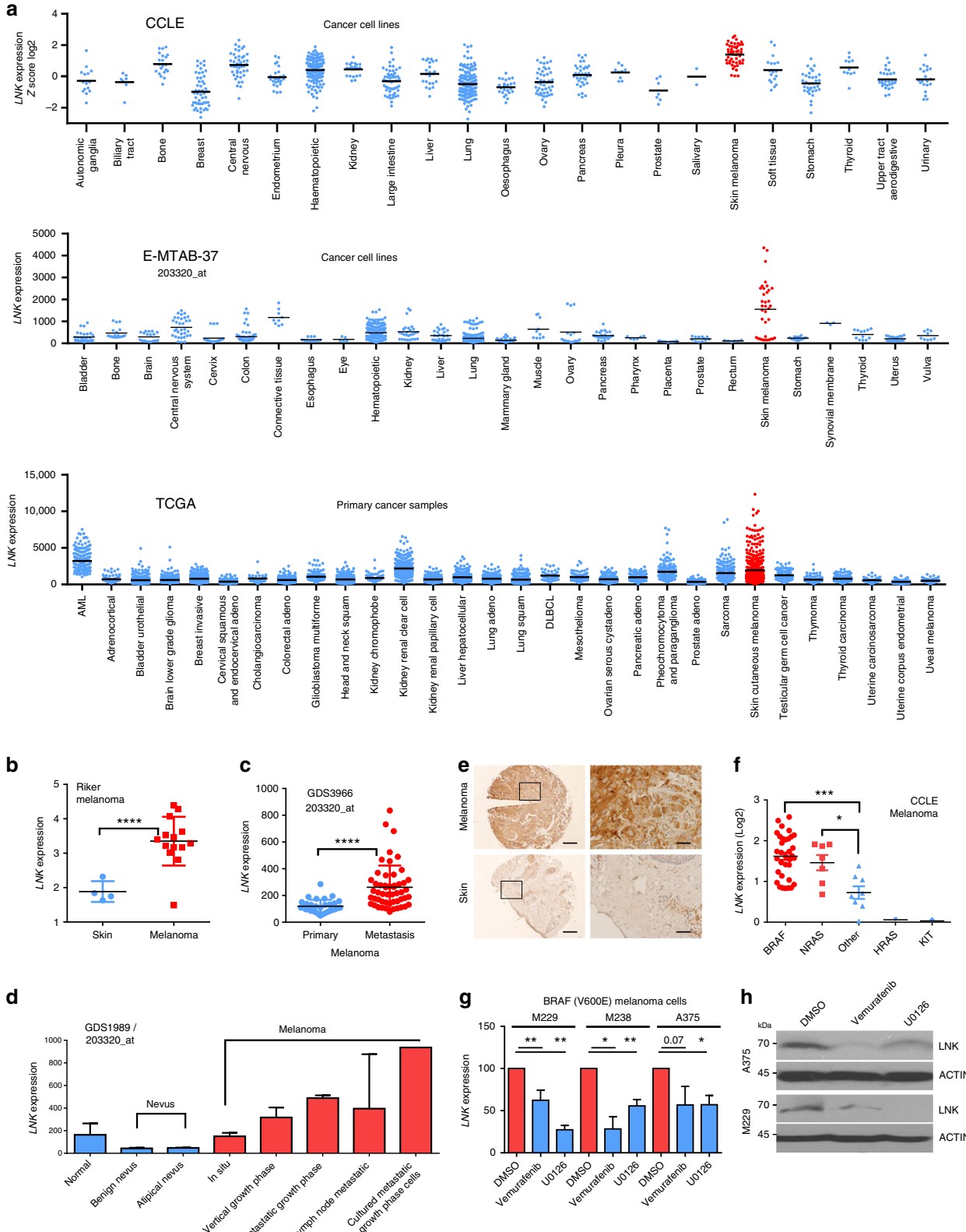

queried the *LNK* mRNA expression and performed a meta-analysis of cDNA microarray data in the GEO database. *LNK* mRNA expression was significantly affected by perturbation of the MAPK signaling in melanoma cells. Inhibition of mutant BRAF (V600E) activity by either the BRAF inhibitor Vemurafenib (PLX4032, Supplementary Fig. 3a) or *BRAF* siRNA markedly reduced the *LNK* transcript levels in melanoma A375

cells (Supplementary Fig. 3b), while forced expression of an activated BRAF (V600E) generated the opposite effect (~four-fold increased *LNK*, Supplementary Fig. 3c). Similarly, silencing expression of mutant NRAS Q61K expression in a doxycycline-inducible murine melanoma model reduced *LNK* mRNA (GSE39984) (Supplementary Fig. 3d). In contrast, forced expression of either activated NRAS Q61K or HRAS/KRAS G12V

**Fig. 1** LNK expression is elevated in melanoma and associated with RAS-RAF-MEK signaling. **a** *LNK* mRNA expression is elevated in melanoma cell lines and primary melanoma samples. Upper panel, *LNK* expression in 877 cancer cell lines (data extracted from microarray data of CCLE). Middle panel, *LNK* expression levels in 332 cancer cell lines (data extracted from microarray data of E-MTAB-37). Lower panel, *LNK* expression in primary cancer samples (data retrieved from TCGA RNA sequencing data using Cbio cancer portal). **b** *LNK* expression is elevated in melanoma, compared with normal skin tissue (data obtained from Oncomine database). Mean ± (+ & −) SD, ****$p < 0.0001$, unpaired *t*-test. **c** *LNK* expression is upregulated in advanced melanoma (metastasis) compared with primary melanoma. Mean ± SD, ****$p < 0.0001$, unpaired *t*-test. **d** *LNK* expression is upregulated in advanced stages of melanoma (vertical growth phase, metastatic growth phase, etc.) compared with either in situ melanoma or benign nevus. **e** IHC staining of melanoma tissue array. LNK is heavily stained in all of the melanoma tissue cores ($n = 12$), while significantly weaker staining was detected in normal skin controls (skin, $n = 12$). Scale bar = 200 μm (left panel) or 20 μm (right panel). **f** Melanoma cell lines in CCLE were classified into different groups based on their oncogenic drivers (BRAF V600E, NRAS Q61K/L, HRAS, c-KIT and other drivers). *LNK* expression levels were significantly elevated in cell lines driven by mutant BRAF V600E and NRAS Q61K. *$p < 0.05$; ***$p < 0.001$; unpaired *t*-test. **g**, **h** Inhibitor of BRAF [Vemurafenib (PLX4032)] or MEK (U0126) consistently downregulated LNK expression in three melanoma cell lines carrying the BRAF V600E mutation. Real time PCR was performed 24 h after exposure to the inhibitors (**g**, mean + SD, $n = 3$). Western blot was performed 48 h after the treatment (**h**). *$p < 0.05$; **$p < 0.01$; unpaired *t*-test. Error bars represent SD. All $p$ values were calculated using two tailed *t*-test

in the immortalized melanocyte cell line Mel-ST significantly increased *LNK* transcripts (GSE62827, Supplementary Fig. 3e). Suppressing the MAPK kinase activity in melanoma cells with a MEK inhibitor (U0126) consistently downregulated *LNK* expression (Supplementary Figs. 3b, d). We performed RT-PCR and western blot to validate our in silico observations. In three melanoma cell lines that harbored the BRAF V600E mutation (A375, M229 and M238), treatment of the cells (24 h) with either the BRAF inhibitor vemurafenib (PLX4032) or a MEK inhibitor (U0126) significantly downregulated the *LNK* mRNA and protein expression (Fig. 1g, h). Collectively, these data suggest that aberrant expression of LNK is correlated with the hyperactivated RAF/MAPK signaling in melanoma.

**LNK promotes growth and survival of melanoma cells**. To evaluate the biological relevance of elevated LNK in melanoma cells, we first examined the effect of LNK silencing in melanoma cells using the database of Novartis DRIVE cancer cell lines [https://oncologynibr.shinyapps.io/drive/]. In this large-scale (398 different cancer cell lines) shRNA library screening project, cells were infected with shRNA library pool targeting 7837 genes (each gene was targeted by ~20 different shRNAs)[34]. The lentiviral shRNA infected cells were allowed to grow for 14 days after infection. The presence of each shRNA was analyzed by high-throughput-sequencing. Enrichment of shRNA at day 14 compared to day 0 indicated that it targeted an anti-growth gene, while depletion of a shRNA suggested it targeted a pro-growth gene (e.g., shRNAs targeting *BRAF* in melanoma, Supplementary Fig. 4). In 27 of the 35 melanoma cell lines examined, *LNK* shRNA (ATARiS value) was consistently decreased at day 14 (compared with day 0), suggesting that silencing LNK by shRNA retards cell growth of most melanoma cell lines (Fig. 2a).

We generated a number of melanoma cell lines that stabilized either over-expressed (OE) or silenced LNK (either by shRNA or CRISPR-Cas9). Lentivirus-transduced melanoma cells were selected using puromycin, and either forced-expression or silencing of LNK were confirmed by western blot. By MTT assay, silencing LNK modestly retarded growth of melanoma cell lines, while forced expression of LNK showed either no significant difference (A375, Supplementary Fig. 5) or modestly reduced cell growth (M202). However, forced expression of LNK enhanced the anchorage-independent clonal growth in soft agar (Fig. 2b) and generated bigger tumors in an in vivo xenograft model (Fig. 2c, d); while silencing of Lnk using CRISPR-Cas9 reduced the tumor formation of murine melanoma B16/F10 cells (Fig. 2c, d). These data suggest that LNK modestly enhances tumorigenesis and the self-renewal potential of melanoma cells.

In addtion, LNK enhances cell survival in various adverse conditions. Melanoma cells with enforced expression of LNK

showed increased resistance to anokisis (induced by cell growth in ultra-low attachment surfaces resulting in cells detaching from their surrounding extracellular matrix) (Fig. 2e, f). Indeed, western blot analysis of these cells showed a reduction of expression of both major apoptotic markers (cleavage caspase 3/9, cleavage PARP) and pro-apoptotic BH3 protein BIM (Fig. 2g), suggesting resistance to apoptosis in LNK-overexpressing cells. Similarly, overexpression of LNK in melanoma cells protected them from cell death when grown in either nutrient-deprived media [100% PBS or 1:10 diluted RPMI (with 90% PBS)] or with an inhibitor of transcription (Actinomycin-D, Fig. 2h).

**LNK inhibits signaling of IFN-STAT1**. Recent studies suggest that the IFN-JAK1/2-STAT1 pathway plays a central role in T cell-mediated killing of tumor cells; and perturbation of this pathway leads to resistance of immune check point blockade in melanoma[15,16]. IFN released by CD8+ T cells binds to the IFN receptors of tumor cells, stimulating expression of a number of IFN responsive genes signaling through JAK-STAT1 pathway. LNK is a well-characterized JAK-STAT suppressor in hematologic malignancies[20,21], prompting us to test whether the prominently expressed LNK in melanoma cells was involved in regulation of the IFN pathway. Melanoma cells exposed to recombinant IFN protein [either type I (alpha, beta) or type II (gamma)] quickly activated the interferon signaling pathway, as evidenced by heavy phosphorylation of STAT1 as early as 30 min following IFN treatment (Fig. 3a, b). Activated STAT1 migrated into the nucleus initiating transcription of downstream interferon regulatory/responsive genes. Forced expression of LNK profoundly suppressed the IFN (either alpha, beta or gamma) induced phosphorylation of STAT1 (Fig. 3), while silencing of LNK using either shRNA or CRISPR-Cas9, generated the opposite effect. Consistently, the major downstream markers of IFN signaling, IRF1 and PD-L1, were downregulated in the LNK overexpressed cells, while higher levels of these proteins were detected in the LNK silenced cells. To preclude the possibility that fetal bovine serum (contains different cytokines/growth factors and may affect LNK expression and downstream signaling) affects our analysis, experiments were performed either with or without serum using several cell lines. Results were consistent with LNK regulating IFN-JAK-STAT in both culture conditions.

We performed immunoprecipitation (IP) to examine the protein interaction between LNK and other proteins. Melanoma cells were treated with IFN gamma for 30 min, lysed and LNK protein was pulled-down using LNK antibody (Santa Cruz (A-12): sc-393709). Western blot analysis showed that STAT1 was pulled down together with LNK protein (Fig. 3e). Similarly, reciprocal immunoprecipitation using STAT1 antibody (Cell Signaling, 9172S) showed LNK was co-precipitated with STAT1 protein. We performed GST-pull down experiments to examine

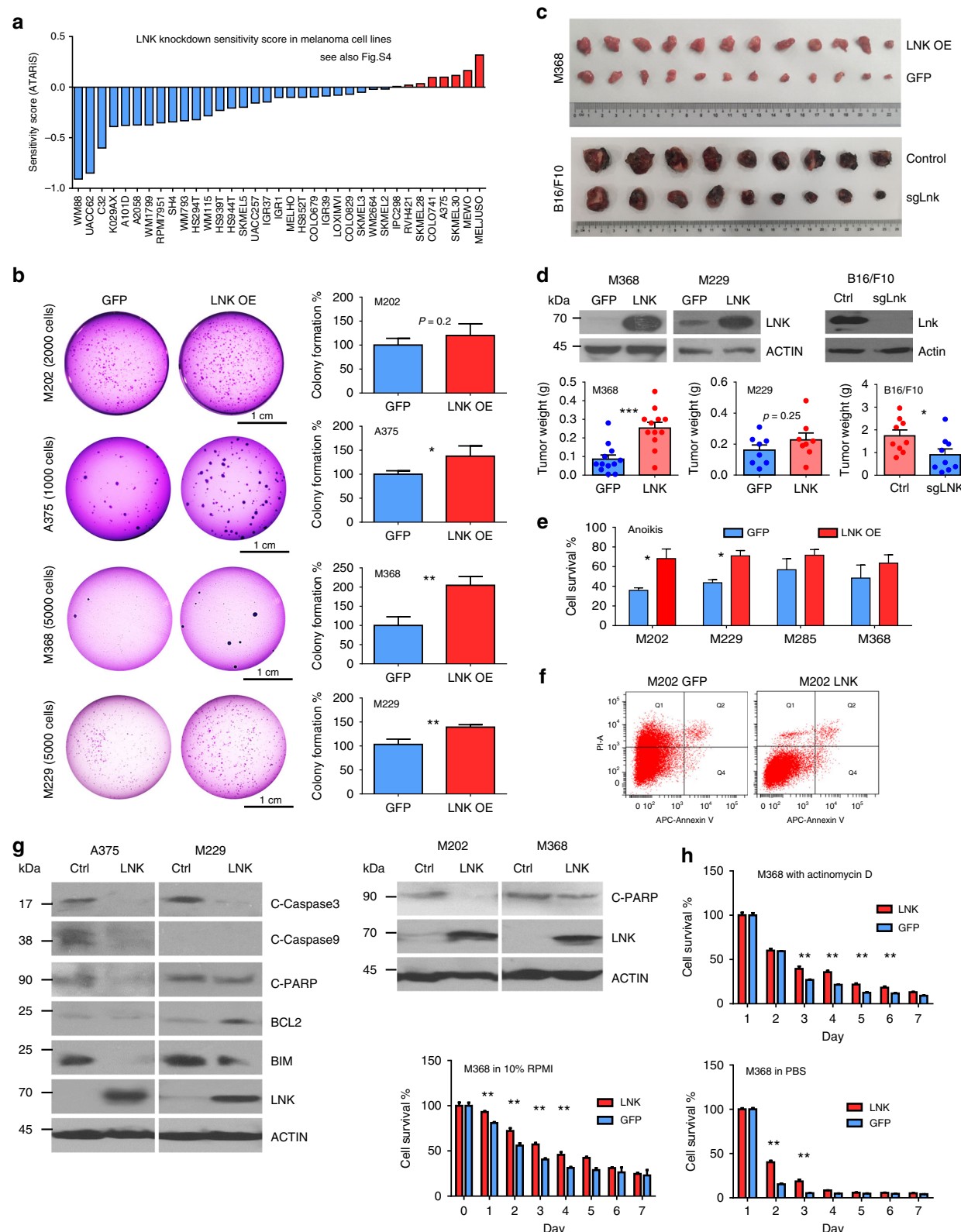

whether a direct interaction occur between LNK and STAT1 protein. Recombinant GST fusion protein of LNK (PH or SH2 domain) and STAT1 were generated and purified using *E.coli* strain TXK1 (Stratagene). The recombinant proteins were incubated with the cell lysate of HEK293T cells transfected with Flag tag STAT1 expressing constructs (GST-LNK pull down experiment) or cell lysis of A375 expressing V5-LNK (GST-STAT1 pull down experiment). Both PH and SH2 domain of LNK, but not the GST control protein, can bind to the STAT1 protein and they were pulled down together as a protein complex (Fig. 3f, g). As an SH2 domain adaptor protein, LNK may interact with STAT1 helping to recruit phosphatase(s) to dephosphorylate the STAT1 protein. We are currently performing SILAC (Stable Isotope Labeling by/with Amino acids in Cell culture) mass

**Fig. 2** LNK promotes cellular growth and survival. **a** ATARiS profiles of *LNK* shRNA in Novartis cancer cell lines shRNA library database [https://oncologynibr.shinyapps.io/drive/]. *LNK* shRNA (~20 different shRNA targeting different region of *LNK*) were depleted/decreased in most of the melanoma cell lines (blue color) on day 14 post-transduction. **b** LNK promotes anchorage independent growth of melanoma cells. Left panel of each picture shows clonal growth in soft agar, while the corresponding bar graph (right panel) enumerates colony formation (mean + SD, $n = 4$). GFP, overexpressing GFP control; LNK OE, overexpression of LNK. *$p < 0.05$; **$p < 0.01$; unpaired $t$-test. **c** Photographs of tumor formation of murine xenografts. Upper panel, M368 xenografts grown in NSG mice show that LNK overexpressing cells formed larger tumors compared with GFP controls. Lower panel, murine melanoma B16/F10 xenograft results after Lnk silencing by CRISPR-Cas9. **d** LNK promotes tumorigenesis in murine xenograft model. Left and middle panels, bar graphs show tumor growth of melanoma cells overexpressing LNK (M368 and M229). Right panel, bar graphs show tumor growth of B16/F10 with silencing expression of LNK. Mean + SEM. *$p < 0.05$; ***$p < 0.001$; unpaired $t$-test. **e, f** Force expression of LNK (LNK OE) enhances cell survival and confers resistance to anoikis related cell death (induced when cells are grown anchorage-free). *$p < 0.05$, unpaired $t$-test. **f** Representative result of Annexin V staining after melanoma cells was grown on ultra-low binding plates. GFP, overexpression GFP control; LNK, overexpression of LNK. **g** Overexpression of LNK attenuates anoikis cell death. Cells were grown on ultra-low attachment plates for 48 h to induce anoikis before protein extraction. The western blot showed that overexpression of LNK reduced expression of apoptosis markers. Ctrl, control cells; LNK, cells overexpressing LNK. **h** LNK enhances cells survival in adverse conditions: M368 melanoma cells were grown either in nutrition deprived media (100% PBS or 1:10 diluted DMEM media, diluted with isotonic PBS) or exposed to transcription inhibitor actinomycin-D (mean + SD, $n = 3$). **$p < 0.05$, unpaired $t$-test. Error bars represent either SEM (**d**) or SD (**b**, **e**, and **h**). All $p$ values were calculated using two tailed $t$-test

spectrometry experiments to identify the downstream phosphatase involved in this regulatory process.

RNA sequencing and cDNA microarray comprehensive profiling were performed to identify genes whose expression changed using melanoma cell lines A375 and M202 [overexpressed or silencing LNK± (either with or without) exposure to IFN-γ for 24 h]. In the A375 cell line, 394 genes were downregulated and 1306 genes were upregulated in the LNK-overexpressing cells vs control cells (with a cutoff of two-fold) treated with IFN-γ. Pathway analysis of upregulated genes showed over-representation of cell cycle progression genes in LNK-overexpressing cells treated with IFN-γ (e.g., upregulation of MYC and CCNE2), while many genes involved in autoimmune disease were downregulated in these cells (pathways related to inflammatory bowel disease, graft versus host disease, Fig. 4a, left panel,+IFN-γ 24 h). In the same cells cultured without IFN-γ, forced expression of LNK upregulated genes of the NF-KB signaling, and down-regulated genes involved in the extracellular matrix (ECM) interaction pathway (Fig. 4a, right panel, without IFN-γ). As anticipated, expression of a series of interferon-inducible genes [e.g., IRF1, IFI27 (promote cell death, mediate the IFN induced apoptosis), IFI6, IFI35, ISG20], as well as IFN downstream effectors [e.g., IDO1 etc.] were markedly upregulated following IFN-γ treatment of control melanoma cells; this upregulation was significantly attenuated in cells overexpressing LNK [both A375 (Fig. 4b) and M202 (Supplementary Fig. 6)]. Conversely, silencing of LNK significantly enhanced expression of interferon-inducible genes (Fig. 5, Supplementary Fig. 7). Analysis of the RNA sequencing data using Gene Set Enrichment Analysis (GSEA) of cells silencing LNK showed a remarkable enrichment of the expression signatures of interferon (False Discovery Rate (FDR) $q$-values = 0), graft versus host disease, allograft rejection, immune effector process and antigen processing pathway (Fig. 5a). An inverse correlation (negative enrichment) of MYC targets and telomeres signature, as well as melanoma relapse gene expression signature were also observed (signature enriched in control cells, Supplementary Fig. 8). Depletion of Myc expression has recently been shown to reverse the immune evasion of lung cancer cells[35]. Notably, expression of a number of HLA genes was suppressed by LNK. For example, a series of MHC class I and II genes, including HLA-DMA, HLA-DOB, etc, were downregulated in the LNK overexpressing A375 cells (Figs. 4b, 5b) and significantly upregulated in the same cell line silenced with LNK using CRISPR-Cas9 guide RNA. Hence, prominent expression of LNK down-regulates the IFN-induced MHC gene expression, probably leading to a decreased MHC-mediated antigen presentation and potentially reducing T cell recognition.

Recombinant IFN alpha has been used as an adjuvant therapeutic reagent for resected, advanced stage melanoma;[36] and IFN can directly suppress proliferation of melanoma cells and induce their apoptosis[37]. Indeed, melanoma cells treated with interferon gamma have markedly suppressed growth (Fig. 6a). Overexpression of LNK attenuates this effect and confers a selective growth/survival advantage (Fig. 6a, b, MTT and foci formation assays). In contrast, silencing of LNK by either shRNA or CRISPR-Cas9 potentiates the effect of interferon, reducing cell proliferation and formation of foci. Interferon gamma-induced rapid apoptosis, reflected by caspase-mediated cleavage of Poly (ADP-ribose) polymerase (PARP) after IFN exposure. Forced expression of LNK potently inhibited cleavage of PARP and suppressed IFN induced apoptosis (Fig. 3a, d). The A375 cell line is resistant to interferon induced apoptosis[38], and cleavage of PARP was hardly detectable, but a marked retardation of cell growth was observed upon IFN-γ treatment. The anti-proliferative effect of IFN in this cell line was probably due to the suppression of cell cycle progression; and as suggested from our RNA sequencing data, LNK probably reversed this effect through upregulation of cell cycle pathway genes (Fig. 4a).

Based on the above observations, we hypothesized that a higher level of LNK expression will suppress the interferon pathway and confer a selective survival advantage to melanoma cells; therefore, those patients who have higher levels of LNK expressed in their tumor cells will be less likely to respond to PD-1 antibody therapy. We examined the *LNK* mRNA expression using RNA sequencing data of a cohort of melanoma patients who were uniformly treated with PD-1 antibody (data retrieved from GSE78220)[39]. RNA was extracted from melanoma biopsy samples before treatment. A trend towards higher levels of *LNK* mRNA was found in non-responding patients [responding cohort ($n = 15$), FPKM mean value = 15.89; non-responding cohort ($n = 13$) = 32.43], although the difference did not reach statistical significance ($p = 0.13$, unpaired $t$-test) (Fig. 6c). In contrast, the mRNA level of the other two SH2B family members (*SH2B1* and *SH2B2*) showed no difference in expression between the responding and non-responding patients (Supplementary Fig. 9). To test further this hypothesis, murine melanoma cells (B16/F10 or D4M.3A) with silenced Lnk were injected into syngeneic C57BL/6 mice, and the mice were treated with anti-mouse PD-1 antibody (Fig. 6d–f, Supplementary Fig. 10). Silencing Lnk by CRISPR-Cas9 enhanced the tumor-suppressive effect of PD-1 antibody retarding tumor growth, supporting the concept of further exploring the therapeutic potential of targeting LNK plus immune checkpoint therapy.

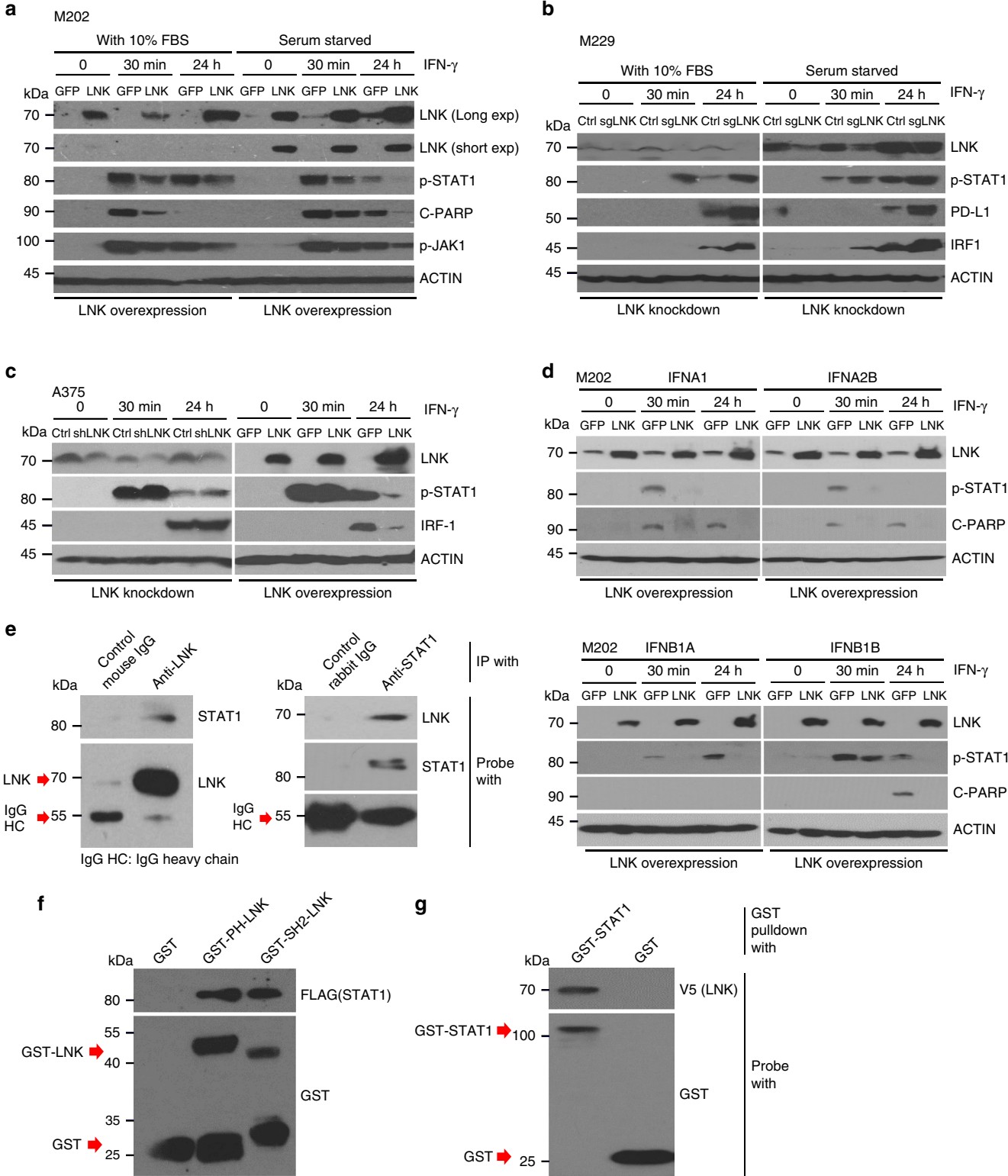

**IFN-STAT1 induces LNK expression as a negative feedback**. Melanoma cells treated with IFN-γ upregulated LNK expression (Fig. 6g). Analysis of the ChIP-seq data from the ENCODE database [https://www.encodeproject.org/] reveals strong IRF-1 enrichment at the transcriptional start site of *LNK* associated with IFN-γ treatment (Fig. 6h, 6 h), suggesting that activated IFN-STAT1 signaling turns on the transcription factor IRF-1 to activate the transcription of *LNK*. This upregulation may act as a negative feedback mechanism to prevent over-activation of the pathway and fine tune the magnitude and duration of signaling of IFN. Melanoma cells may also use this signaling axis to ameliorate the anti-proliferative effect of IFN.

## Discussion

Melanoma harbors the highest mutational burden among cancers[40] and is frequently infiltrated with T lymphocytes[41]. Their high mutational load increases the likelihood of tumor specific

**Fig. 3** LNK suppresses signaling of the IFN-STAT pathway. **a** Western blots show forced expression of LNK suppress the IFN-γ (2000 U/ml, 24 h) induced STAT1 phosphorylation in M202 cell. Cells were growth in either normal complete media (with 10% FBS) or serum starved (24 h) before IFN-γ treatment. GFP, GFP overexpressing control; LNK, LNK overexpressing cells; C-PARP, cleavage PARP; Exp, exposure time. **b** Western blots show silencing of LNK enhances the IFN-γ (2000 U/ml, 24 h) induced STAT1 phosphorylation and enhanced expression of downstream markers (IRF-1 and PD-L1) in M229 cell. Cell were grown in either normal complete (with 10% FBS) or serum starved media 24 h before the IFN-γ treatment. Ctrl, control; sgLNK, CRISPR-Cas9 guide RNA targeting *LNK* (sgLNK-1). **c** Western blots showed that forced expression of LNK suppresses the IFN-γ induced STAT1 phosphorylation and IRF1 expression in A375 cells (left panel), while silencing LNK generated the opposite effect. Ctrl, non-target shRNA control; shLNK, shRNA targeting *LNK* (shLNK-16). **d** Western blots show force expression of LNK suppresses interferon alpha (IFNA1 as well as IFNA2B) or interferon beta (B1A, Interferon beta 1A; B1B, Interferon beta 1B) induced STAT1 phosphorylation in M202 cell. **e** Immunoprecipitation (IP) experiments showed that LNK interacted with STAT1 protein. Melanoma cells were treated with IFN gamma (2000 U/ml) for 30 min. Lysates were extracted (m-PER protein extraction solution) and proteins were pull-downed using LNK (Santa Cruz Biotechnology (A-12): sc-393709) or STAT1 (Cell Signaling, 9172) antibody. **f** GST-pull down experiments showed that the PH and SH2 domains of LNK bind to STAT1 protein. GST-PH-LNK, GST protein fused to the PH domain of LNK; GST-SH2-LNK, GST protein fused to the SH2 domain of LNK. **g** GST-pull down experiments showed the GST-STAT1 protein (expressed from *E.coli*) directly binds to the LNK protein (V5 tag)

neoantigen (encoded by the mutant protein) expression on the surface of melanoma cells which can be recognized by T cells[42,43]. Melanoma cells can develop different approaches to evade immune surveillance[12] including: (1) upregulation of the cell surface expression of T cell inhibitory molecules (e.g., PD-L1, TIM3, etc[44,45].); (2) downregulation of IFN signaling (e.g., loss of function mutation of JAK2[15,46]); (3) suppression of presentation of neoantigens on the tumor cell surface to reduce T cell recognition [e.g., mutation/deletion or downregulation of the molecules involved in antigen processing in the endoplasmic reticulum (ER)[12] (e.g., TAP etc.) or antigen surface presentation (e.g., MHC or β2-microglobulin (B2M) protein[15]; (4) recruitment of immuno-suppressive cells [e.g., myeloid-derived suppressor cells (MDSCs)] or regulatory T cells directed to the tumor micro-environment[47]. Among these escape mechanisms, IFN signaling plays a central role in coordinating T cell activity and killing cancer cells. In a murine model of melanoma, blockade of Ifna signaling using an anti-Ifnar antibody completely abolish tumor rejection mediated by Pd-1 antibody therapy[48]. IFN released by CD8+ T cells activates the JAK-STAT1 signaling pathway; upon activation, the IFN receptor recruits JAK1/2 tyrosine kinase to phosphorylate the major downstream effector STAT1. The latter translocates into the nucleus to stimulate transcription of INF-stimulated response genes. We identified LNK as a negative regulator of the signaling by JAK-STAT in melanoma. Elevated LNK protein in these melanoma cells confers selective survival advantage by suppressing signaling by IFN. The elevated LNK is associated with the mutant hyper-activated RAS-RAF signaling pathway in melanoma cells. Hence, a potential therapeutic rationale for the combination of a RAF/MEK inhibitor (down-regulate LNK expression) with an immune checkpoint blockade antibody[49,50].

The observation that LNK suppresses the IFN-induced MHC class II and I genes is of interest. The MHC complex mediates antigen cell surface presentation and plays a central role in neoantigen-mediated T cell recognition[51]. Recently, Johnson et al found that MHC-II gene expression was correlated with response to PD-1/PD-L1 antibody therapy[52] and the expression of HLA-DR correlated with pro-immune/inflammatory gene signature[52]. Response to PD-1 antibody therapy was associated with pre-existing IFN-γ signaling including the display of MHC class II proteins on untreated melanoma patient samples[53]. Overexpression of LNK suppresses expression of MHC genes, leading to a reduction of cellular antigenicity, helping the tumor cells evade immune surveillance. Indeed, a negative correlation of LNK and MHC gene expression was observed in melanoma samples (Supplementary Fig. 11). Notably, recent genome-wide association studies (GWAS) identified a single nucleotide polymorphism (SNP) in the LNK coding region which is significantly associated with a variety of autoimmune diseases [e.g., celiac disease[54],

autoimmune hepatitis type 1[55]]. LNK may suppress pro-inflammatory cytokines induced by autoantigen presentation through down-regulation of MHC/HLA genes in these diseases, and this deserves further study.

The clinical translation of targeting LNK in melanoma patients may have the potential to improve the effect of immune check-point therapy and increase patient survival. Pharmacologic targeting LNK using small molecule, if feasible, not only may enhance the tumor immunity in cancer cells, but may also enhance the activity and cell proliferation of tumor infiltrating T cells. Silencing LNK using small molecules may also enhance the JAK-STAT signaling in lymphocytes enhancing their proliferation and activation. Targeting LNK's downstream phosphatase may represent an alternative approach to enhance the JAK-STAT signaling and tumor immunity in cancer cells[18]. We are currently performing mass spectrometry experiments to identify this phosphatase.

In summary, we showed that expression of the adaptor protein LNK is significantly elevated in melanoma cells, and enhanced transcription of *LNK* is associated with the signaling of the hyperactivated RAS-RAF-MEK pathway (Fig. 6i). Ectopic expression of LNK enhances cell survival and tumor growth and suppresses IFN-induced apoptosis/cell cycle arrest. Our study uncovered an unappreciated function of LNK in melanoma and underscores the important role of IFN-STAT1-LNK signaling in this potentially devastating disease.

## Methods

**Study design**. For the murine xenograft experiment, a sample size of $n \geq 6$ per group was used to achieve a statistical significance of $p < 0.05$. All tumor samples were included in the analysis. Experiments were performed in a non-blinded way.

**Statistical analysis**. GraphPad Prism 6 was used for the statistical analysis. Data were analyzed using unpaired two-tailed t test and presented as means ± SD, unless otherwise indicated. $P < 0.05$ was considered to be statistically significant.

**Cell lines**. Melanoma cell line A375 and B16/F10 were from ATCC; primary human melanoma cells (M202, M229, M238, M285, M368) were kindly provided by Dr Antoni Ribas, UCLA. Murine melanoma cell line D4M.3A was provided by Dr David W.Mullins. Cells were maintained in DMEM medium with 10% fetal bovine serum (FBS) and 1% Penicillin-Streptomycin. HEK293T cells were cultured in DMEM medium with 10% FBS.

**Reagents**. Recombinant human Interferon alpha 1 (CYT-291), Beta 1a (CYT-236), Beta 1b (CYT-234) Gamma (IFN-γ, CYT-206) were obtained from Prospecbio.

**Immnuohistochemical (IHC) staining**. Melanoma tissue array (ME242a, obtained from US Biomax Inc.) was stained with LNK antibody (R&D, AF5888, the specificity and titration of the antibody for IHC experiment was tested with positive and negative controls[56]). Staining of tissue arrays of 163 melanoma patient samples was performed at Melanoma Institute Australia, Sydney, Australia.

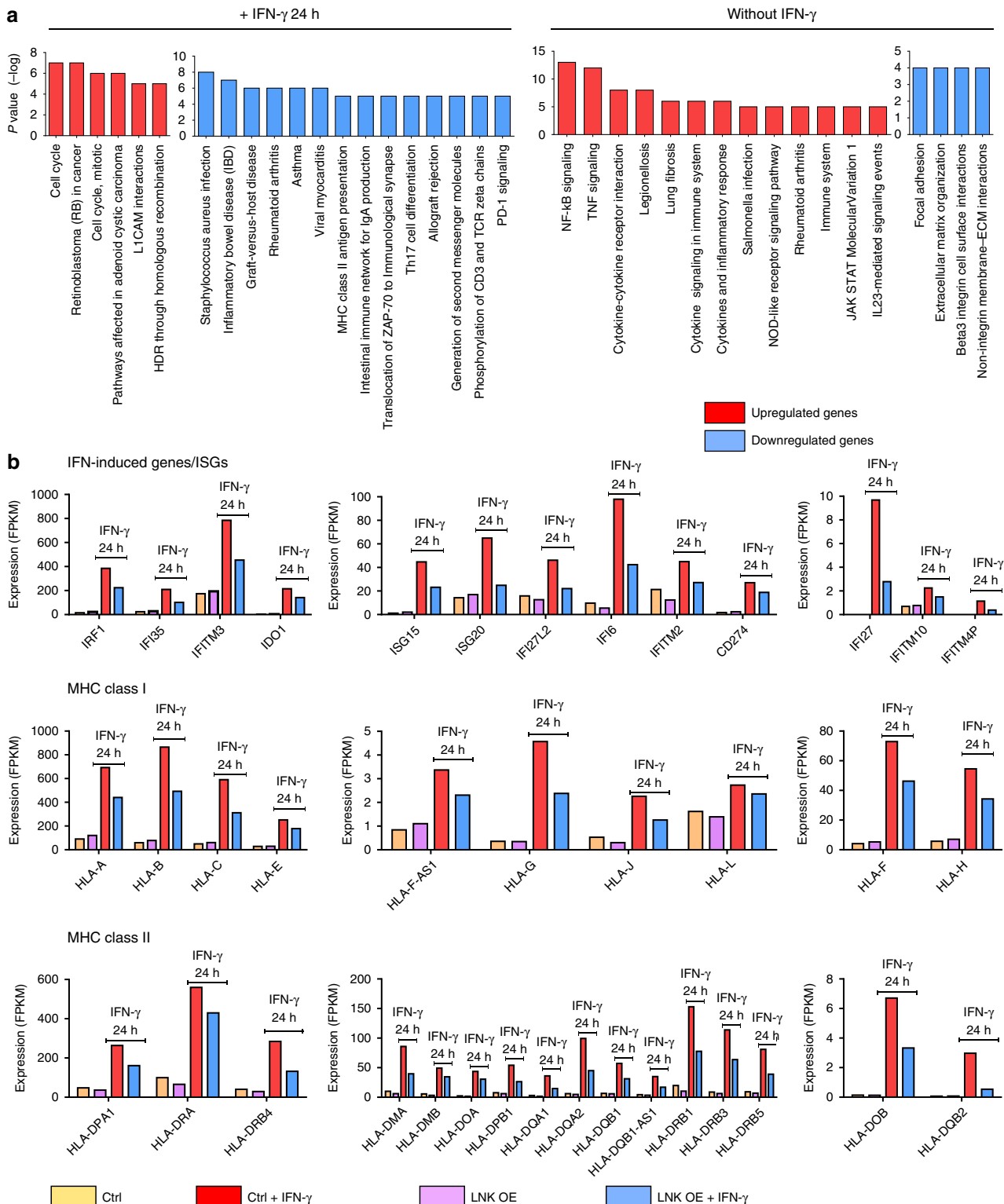

**Fig. 4** LNK attenuated the IFN induced gene expression. **a** Pathway enrichment analysis: A375 cells with forced expression of LNK (LNK OE) were cultured either with (left panel) or without (right panel) IFN-γ (2000 U/ml). Enriched pathways of either upregulated (red color) or downregulated (blue color) genes in forced LNK expressing A375 melanoma cells are displayed with their *p* value. Pathway analysis was performed using [http://cpdb.molgen.mpg. de/]. **b** LNK attenuated the IFN-γ (2000 U/ml, 24 h) induced gene expression in A375 melanoma cells. Upper panel, LNK attenuated expression of IFN-γ interferon response genes. Middle panel, LNK attenuated IFN-γ induced expression of MHC class I genes. Lower panel, LNK attenuated IFN-γ induced expression of MHC class II genes

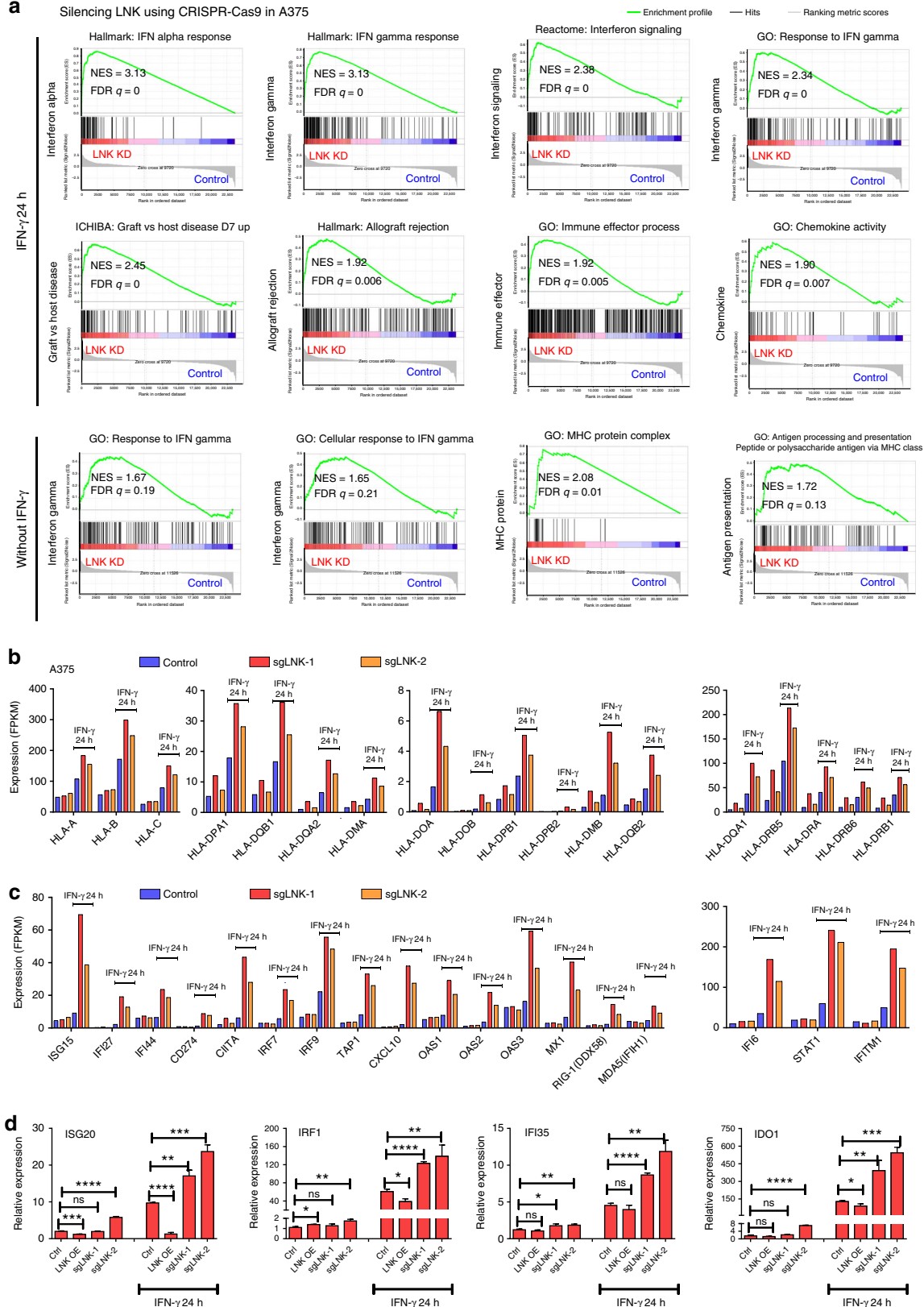

**Real time qRT-PCR**. Total RNA was extracted from cells using RNeasy Mini Kit (QIAGEN). *LNK* mRNA expression was determined by real time PCR using primer DLW13 CACTTTCCCTCGGTCGTG and DLW 14 GGGACAGCCAGAA-GAACTAA (targeting exons 7 and 8 including the intervening intron of human *LNK* gene). Primers of IFN induced genes are listed in Supplementary Table 3. β-Actin or GAPDH were used for normalization.

**Lentivirus and generation of stable cell lines**. Lentiviral CMV-GFP plasmid SHC003 was obtained from Sigma-Aldrich. Lentivirus construct to express LNK was generated by replacing the GFP with LNK coding region[56]. For gene silencing, shRNA plasmids targeted to *LNK* [TRCN0000265715 (shRNA15), TRCN0000265716 (shRNA16), TRCN0000256095 (shRNA95) and Non-targeting shRNA SHC002 were obtained from Sigma-Aldrich. CRISPR-Cas9 sgRNAs were

**Fig. 5** Silencing of LNK enhances IFN signaling in melanoma cells. **a** GSEA analysis of significantly enriched gene expression signatures in A375 melanoma cells with silenced LNK using different CRISPR-Cas9 guide RNA. Cells were either cultured with or without IFN gamma (400 U/ml, 24 h); and gene expression was analyzed using RNA sequencing. NES, normalized enrichment score; FDR, false discovery rate. The color scale indicates the positive (red) or negative (blue) correlation. **b** Silencing of LNK enhanced the IFN-γ induced (400 U/ml, 24 h) gene expression of MHC class I and II. **c** Silencing of LNK enhanced the IFN response gene expression in A375 cells. **d** Real time PCR analysis of IFN induced genes expression in A375 cells with either forced expression (LNK OE) or silencing of LNK (sgLNK), either with or without IFN-γ treatment (2000 U/ml, 24 h). *$P < 0.05$; **$P < 0.01$; ***$P < 0.001$; ****$P < 0.0001$; ns, not significant; unpaired two tailed t-test. Error bars represent SD

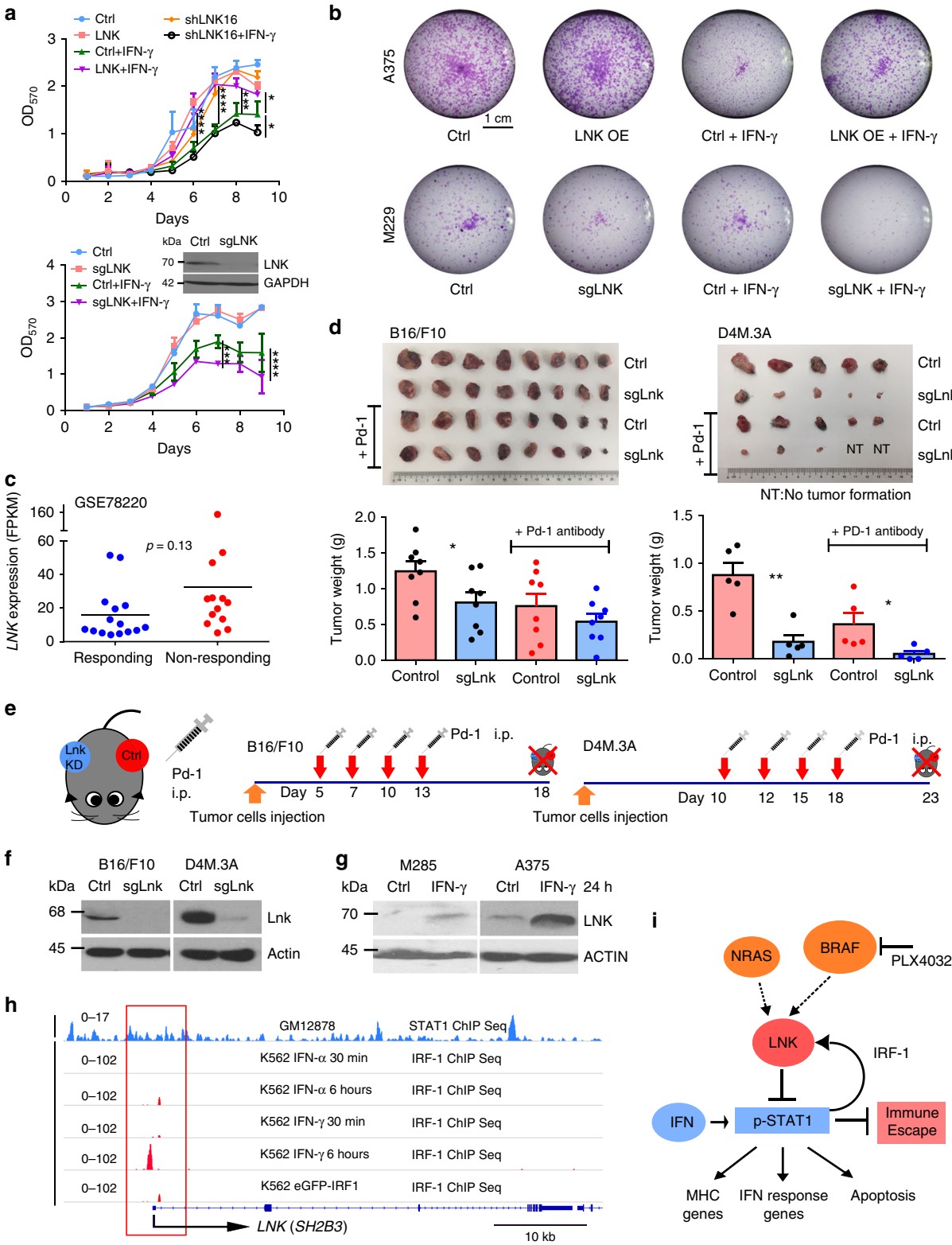

**Fig. 6** IFN induces expression of LNK as a negative feedback. **a** MTT assay of A375 melanoma cells with either force expressing or silencing of LNK either in the presence or absence of IFN-γ (2000 U/ml). Upper panel, silenced LNK using shRNA (shLNK16). Lower panel, silenced LNK using CRISPR-Cas9 sgRNA (sgLNK). Ctrl, control A375 cells; LNK, LNK overexpressing cells. Mean ± SD; ***$P < 0.001$; ****$P < 0.0001$; Two way ANOVA. **b** Foci formation assays of melanoma cell lines with either force expression of LNK (LNK OE) or silencing of LNK (sgLNK), either with or without IFN-γ exposure. IFN-γ (2000 U/ml) was added 2 h after seeding the cells. Ctrl, control. **c** Higher levels of *LNK* expression occurred in pretreatment biopsy samples of melanoma patients who did not respond to PD-1 antibody therapy. Data were retrieved from GEO dataset GSE78220, unpaired two tailed *t*-test. **d**, **e** Silencing Lnk enhanced tumor suppressive effect of Pd-1 antibody. Control (control CRISPR-Cas9 vector) and Lnk silenced (CRISPR-Cas9 with sgLNK guide RNA) B16/F10 (left panel, $n = 8$) or D4M.3 A cells (right panel, $n = 6$. Two mice did not growth any tumor at day 10 were excluded before separation of the cohorts) were subcutaneously injected into both flanks of the same mice (8–12-weeks-old syngeneic C57BL/6 mice). Mean + SEM; *$P < 0.05$; **$P < 0.01$; unpaired two tailed *t*-test. **f** Western blot analysis showed Lnk silencing in either B16/F10 or D4M.3 A cells. **g** Exposure to IFN-γ induced LNK expression in melanoma cells. Melanoma cells were treated with IFN-γ (2000 U/ml, 24 h) and expression of LNK was examined by western blot. Ctrl, control. **h** ChIP-seq binding pattern of STAT1 and IRF-1 at transcription start site of *LNK*. Data were retrieved from ENCODE project database. Number of each panel reflects scale of ChIP enrichment. **i** Diagram shows the IFN-STAT1-LNK signaling axis in melanoma cells. Error bars represent SEM (**d**) or SD (**a**)

---

generated based on the pLENTI-CRISPRv2 according to the procedure described in [http://www.addgene.org/crispr/zhang/]. The sequences of all the shRNA and shRNA constructs were confirmed by Sanger sequencing using U6 primer. The shRNA and CRISPR-Cas9 sgRNA sequences are listed in Supplementary Table 1.

**Western blot analysis**. Total proteins were extracted using M-PER mammalian Protein Extraction Reagent (Thermo Scientific). The following antibodies were used in this study: anti-human LNK antibody (AF5888, R&D system Inc, 1:2000); anti-mouse/human Lnk/LNK ((A-12) sc-393709, Santa Cruz Biotechnology Inc, 1:500); anti-β-actin (a1978, Sigma-Aldrich, 1:6000); anti-p-JAK2 (Tyr1007/1008, 3776S, Cell Signaling Technology Inc, 1:2500), JAK2 (3230S, Cell Signaling Technology Inc, 1:2500), p-JAK1 (Tyr1022/1023, 3331, Cell Signaling Technology Inc, 1:2500), JAK1 (3344, Cell Signaling Technology Inc, 1:2500), p-STAT1 (Tyr701, 7649, Cell Signaling Technology Inc, 1:2500), STAT1 (9172, Cell Signaling Technology Inc, 1:2500), IRF1 (8478, Cell Signaling Technology Inc, 1:2500), Cleaved Caspase-3 (9661, Cell Signaling Technology Inc, 1:2000), Cleaved Caspase-9 (9501, Cell Signaling Technology Inc, 1:2000), BCL2 (4223, Cell Signaling Technology Inc, 1:2000), BIM (2933, Cell Signaling Technology Inc, 1:2000), c-PARP (Asp214, 5625, Cell Signaling Technology Inc, 1:2500), PD-L1 (13684, Cell Signaling Technology Inc, 1:2000) and GAPDH (2118, Cell Signaling Technology Inc, 1:6000). Detail of all the antibodies are listed in Supplementary Table 2, the source data of western blots are provided in Supplementary Fig. 12.

**Cell proliferation assays**. Cell proliferation assays were performed using MTT. 2000–8000 cells were seeded in 96-well plates and grown in a 37 °C cell incubator. Relative cell numbers were determined by staining the cells with MTT dye at indicated time points. Stained cells were dissolved with SDS-dimethylformamide solution and measured with spectrometer (OD$_{570}$).

**Soft agar colony formation assays**. Cells (1000–5000 cells/well, depending on the cell line, 24 well plates) were resuspended in 2 × DMEM with 20% FBS, mixed 1:1 with 0.7% low-melting agarose and seeded on top of a solid base layer (1% agarose mixed 1:1 with 2 × DMEM with 20% FBS)[57]. Cells were grown at 37 °C until the colonies were visible by eye; these were then stained with 1:50 diluted Gentian Violet and the number of colonies was quantified.

**Murine xenografts**. For human melanoma cell lines, 5–8-weeks-old Nod-SCID mice were used for the xenograft experiments. Indicated number of cells was resuspended in 100 μl of DMEM media [cell line M368 was resuspended in 100 μl FBS together with 100 μl Matrix gel (BD)] and subcutaneously injected into both flanks of the mice. Mice were sacrificed, and tumors were excised at the end of the experiments.

For murine melanoma cell line B16/F10 and D4M.3 A, 8–12-weeks-old syngeneic C57BL/6 mice were used for the experiments. Control (with control CRISPR-Cas9 vector) and Lnk silenced cells (CRISPR-Cas9 with sgLnk guide RNA) were subcutaneously injected into both flanks of the same mice. Mice were randomly separated into two group (either Pd-1 treated or untreated group), 100 μg anti-murine Pd-1 antibody (BE0146, Bio X Cell) was intraperitoneal injected on days 5, 7, 10, 13 (B16/F10 cells) or days 10, 12, 15, 18 (D4M.3A cells) post implantation of the cells (Fig. 6e). Mice were sacrificed, and tumors were excised after 18 (B16/F10) or 23 (D4M.3A) days. The animal study was conducted according to the ethical regulations and was approved (R18-0450) by NUS Institutional Animal Care and Use Committee (IACUC).

**Cell anoikis assays**. Anchor-independent growth was achieved by culturing cells in ultra-low-adherence plates (Costar, Corning, $10^5$ cells/well, and 24-well plates). Cell viability was determined using Vi-CELL viability analyzer (Beckman) at different time points (0, 24, and 48 h). For Annexin V analysis, cells were harvested at 48 hours, stained with Annexin V-APC and Propidium Iodide (BD Biosciences) and examined using flow cytometry (BD LSRII).

**RNA sequencing and microarray analysis**. mRNA expression was profiled using either high-through-put RNA sequencing or Illumina Human HT-12 v4 Expression BeadChip. Real-time RT-PCR was performed to validate the significantly altered genes.

**Reporting summary**. Further information on experimental design is available in the Nature Research Reporting Summary linked to this article.

## Data availability
The RNA sequencing and microarray data has been deposited in the GEO database under the accession code GSE127764, GSE127333 and GSE127965. All the other data supporting the findings of this study are available within the article and its supplementary information files and from the corresponding author upon reasonable request. A reporting summary for this article is available as a Supplementary Information file.

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

## Acknowledgements

We thank Dr. Antoni Ribas in UCLA for generously providing the melanoma cells. Research was supported by the National Research Foundation Singapore under the Singapore Translational Research (STaR) Investigator Award (NMRC/STaR/0021/2014) and administered by the Singapore Ministry of Health's National Medical Research Council (NMRC), the NMRC Centre Grant awarded to National University Cancer Institute of Singapore, the National Research Foundation Singapore and the Singapore Ministry of Education under its Research Centres of Excellence initiatives. This research was also supported by the RNA Biology Center at the Cancer Science Institute of Singapore, NUS, as part of funding under the Singapore Ministry of Education's Tier 3 grants, grant number MOE2014-T3-1-006. R.A.S., J.S.W., and J.F.T. were supported by Melanoma Institute Australia, The University of Sydney Melanoma Foundation, and the Australian National Health and Medical Research Council.

## Author contributions

L-W.D., Q-Y.S., and H.P.K. conceived the project, designed the study and wrote the manuscript. L-W.D., Q-Y.S., L.T.F., X-B.R., and N.V. performed the experiments and analyzed the data. L-W.D., S-Q.Z. and H.Y. performed the in silico data analysis. J.J.E., R. A.S., J.S.W., J.F.T., N.D., J.W.S. performed the IHC staining of melanoma tissue arrays and analyzed the clinical data of the patients, and assisted with final preparation of the manuscript. X-Y.L., J-F.X., M.P., L.X., and D-C.L., assisted with the data analysis. D.W. M. provided the D4M.3A cells.

## Additional information

**Competing interests:** The authors declare no competing interests.

