## [Peer Review File · Nature Communications]

Reviewers' comments:

Reviewer #1, Expertise: SOCS, cytokine signalling (Remarks to the Author):

The manuscript by Ding et al., examines the role of LNK in suppressing IFN gamma signaling in melanoma. LNK has previously been described to regulate JAK/STAT signaling in myeloproliferative disorders and is thought to be a tumor suppressor in this context. Here the authors provide data supporting several key concepts.

- 1) BRAF mutations and NRAS signaling in melanoma increase the expression of LNK and this is associated with a poor outcome for patients.
- 2) LNK is a negative regulator of IFN gamma signaling in melanoma cells, enhancing tumor growth and survival and acting in a negative feedback loop.
- 3) Increased LNK expression is a mechanism for tumors to escape IFN gamma-killing and PD-1 immunotherapy.

This is an important area of research, particularly driven by the new immunotherapies and the need to understand why some patients don't respond and the basis for tumor escape. In addition the data suggests that LNK may be a new immunotherapy target to enhance IFN γ killing and anti-PD-1 therapy.

In general the data support the authors conclusions, however I think two aspects need more work before it is suitable for publication:

1) The Western blots in figure 3 are generally of poor technical quality and need to be improved. They largely show consistent differences with depletion or overexpression of LNK, but the effects of depletion with shRNA or Crispr/cas9 are more variable and less compelling. At minimum a number of panels should be re-run and MW size markers included on all panels: pSTAT1 panel Fig 3D A375 cells, LNK panel Fig 3D A375 cells, all panels in Fig 3E M202 cells. In many of the westerns, samples have spilled into adjacent lanes, reducing confidence in the results and I would suggest that the authors consider the gel % and run the gels more slowly to improve band resolution.

2) PD-1 expression and changes in PD-1 levels with IFN gamma treatment +/- LNK deletion, should be shown for the B16/F10 cell line used in Fig. 6D. This could be done in cell culture to complement the in vivo experiments, although it would be better if expression analysis was performed on excised tumours. Fig. 6D is potentially an important result and although there is a trend towards reduced tumor growth with PD-1 antibody and LNK deletion, it is not significant. I appreciate that the authors may have chosen B16F10 cells as they are only moderately responsive to PD-1 blockade, however it may be necessary to try different cell numbers or a different cell line. Ideally these experiments would have been performed using a BRAF mutant B16 cell line which should have elevated levels of LNK and make the effects of LNK deletion more robust (this would also support the first part of the manuscript). It is important to know how efficient the LNK deletion is in the B16 cells. In addition to tumor weight at the experimental endpoint, the authors could measure tumor size over time.

Other comments:

It is unclear why there is more pSTAT1 in control cells at 24 hours vs 30 min (Fig. 3B)? Usually, Stat phosphorylation peaks within 1 h. Also, there appears to be no reduction in LNK levels with sgLNK at this timepoint in serum starved cells?

The authors should show the level of LNK reduction with overexpression and guide deletion in MC38 and B16/F10 cell lines used in experiment Fig. 2C (also relevant for Fig. 6D)

The authors should comment on why they have performed experiments with and without FBS (Fig. 3).

The authors should check the labeling of "ctrl" vs "GFP" in Fig. 3E.
Typographical error in axis labels "Eexpression" in Fig. 4B.

Reviewer #2, Expertise: LNK, JAK, STAT (Remarks to the Author):

In this manuscript Ding et al presented a very interesting observation that the expression of LNK is elevated in melanoma that is correlated to hyperactive RAS/RAF/MEK signaling. Overexpression of LNK inhibits IFN/SAT1 signaling and reduces the tumor killing effects of IFN. Reversely, silencing LNK expression enhances IFN signaling and potentiates IFN killing effects. Furthermore, they presented evidence showing that LNK expression is upregulated by IFN signaling as a negative feedback mechanism. This work extends the knowledge of LNK inhibition of JAK/STAT signaling in hematopoietic cells, and uncovers a novel role for LNK in melanoma that can be explored for melanoma immunotherapy.

The manuscript performed a comprehensive study of LNK in melanoma. It utilized various public databases to their advantages, multiple cell lines, and more importantly, xenotransplantation of melanoma cell lines manipulated of LNK levels. Although this work is very interesting and potentially of high impact, it is concerning that some statistics are borderline significant or not significant. Furthermore, it heavily relies on overexpression (OE). Knockdown or knockout of LNK have less of a pronounced effect than OE.

1. Figure 1E is a crucial figure for this work, but it used one antibody and no quantification was presented. The authors should validate this antibody with KD/KO. How to normalize and compare IHC intensities from different batches of tissues should be discussed.
2. The correlation between LNK expression levels and RAS/RAF mutation status is interesting, but it is correlative. It is not sufficient to conclude that the RAS/RAF/MEK pathway upregulates LNK expression. As controls, the authors should apply other inhibitors to see if LNK expression is specifically linked to RAS pathway. More importantly, it will be compelling to examine if inhibitors that can reduce cell proliferation, could reduce LNK expression. In another word, LNK expression is correlated with cell proliferation or survival, rather than the RAS/RAF pathway?
3. Figure 2D showed a bit variable results on the effects of LNK OE or KO. Were the tumors that grew similarly to control has less LNK knockdown or knockout? The authors should show the KO/KD efficiency. What does Figure 2E "Anoikis" measure? Figure 2F showed higher% of PI+AnnexinV- cells in the GFP control, that is not consistent with apoptosis. PI+AnnexinV- are simply dead cells or debris, and not accurate for quantification.
4. Western blots in Figure 3 seem to be a bit variable in different cell lines. pSTAT1 and PD-L1 level changes in A375 seem to be inconsistency with the authors hypothesis, or different from other cell lines.
5. Figure 3F, Co-IP experiments showed that LNK interacts with JAK1, pSTAT1, and PTPN2. Are they indirect or direct interactions? Can LNK bind to all of them at the same time as one complex?
6. It would be great to include RNAseq data with sgRNA to LNK, in addition to OE.
7. Figure 5 A, it looks that there are some differences between two different sgRNAs to LNK. sgRNA1 to LNK was not significant in some genes (eg, CD274). The genes in the first row showed a higher percentage change in OE than LNK KO, whereas the second row showed a more pronounced difference in LNK KO than LNK OE. Can authors explain this?
8. Figure 6A lacks statistical analysis.
9. Figure 6D is an important piece of data, but it looks that with PD-1 antibody, there is no significant difference between control and Lnk knockout?
10. The authors showed that IRF-1 may directly bind to LNK promoter and regulate LNK expression. How to explain their data showing that RAS/MAPK pathway upregulates LNK expression?

Reviewer #3, Expertise: melanoma, immune therapy (Remarks to the Author):

Dr. Ding and colleagues suggest a role for LNK in mediating IFN γ signaling and suppressing antitumor immunity. This is a nice contribution although I have a few comments.

Major comments:

1. Several groups have found that MHC-II expression correlates with response to anti-PD-1 (Johnson et al Nat Comm 2016) and tumor intrinsic cell expression of HLA-DR (in similar CLE analysis) correlated with pro-immune gene expression signatures, even in the absence of infiltrating immune cells. Does LNK expression (in CLE or in tumors) correlate with MHC-II (HLA-DR) or MHC-I (e.g. HLA-A)?
2. Are there mutations in LNK in TCGA melanoma samples and if so, are they gain of function/loss of function? Do mutated samples have PDL1, CD8, MHC-I/II expression?
3. Figure 6D - was there any difference in survival in these mice? Any difference in T cell infiltration? At what timepoint were these tumors harvested? Is this a known anti-PD-1 sensitive or resistant model? Just showing differences in tumor size is not very convincing.

Minor comments:

1. Introduction: the rate of durable responses to anti-PD-1 is 30-40% not 20-30%.
2. Similarly, the rate of secondary resistance is closer to 20-33% rather than 40-50%
3. Figures 3C-E - Please make it more clear which panels have forced expression of LNK and which have LNK silenced.
4. Page 11 - should note that adjuvant melanoma treatment has been interferon alpha not interferon gamma

Reviewers' comments:

Reviewer #1, Expertise: SOCS, cytokine signalling (Remarks to the Author):

The manuscript by Ding et al., examines the role of LNK in suppressing IFN gamma signaling in melanoma. LNK has previously been described to regulate JAK/STAT signaling in myeloproliferative disorders and is thought to be a tumor suppressor in this context. Here the authors provide data supporting several key concepts.

- 1) BRAF mutations and NRAS signaling in melanoma increase the expression of LNK and this is associated with a poor outcome for patients.
- 2) LNK is a negative regulator of IFN gamma signaling in melanoma cells, enhancing tumor growth and survival and acting in a negative feedback loop.
- 3) Increased LNK expression is a mechanism for tumors to escape IFN gamma-killing and PD-1 immunotherapy.

This is an important area of research, particularly driven by the new immunotherapies and the need to understand why some patients don't respond and the basis for tumor escape. In addition the data suggests that LNK may be a new immunotherapy target to enhance IFN γ killing and anti-PD-1 therapy.

In general the data support the authors conclusions, however I think two aspects need more work before it is suitable for publication:

1. The Western blots in figure 3 are generally of poor technical quality and need to be improved. They largely show consistent differences with depletion or overexpression of LNK, but the effects of depletion with shRNA or Crispr/cas9 are more variable and less compelling. At minimum a number of panels should be re-run and MW size markers included on all panels: pSTAT1 panel Fig 3D A375 cells, LNK panel Fig 3D A375 cells, all panels in Fig 3E M202 cells. In many of the westerns, samples have spilled into adjacent lanes, reducing confidence in the results and I would suggest that the authors consider the gel % and run the gels more slowly to improve band resolution.

Thank you very much for the suggestions of gel percentage and running condition. According to the suggestions, we repeated the experiments and replaced the low quality photos of Fig. 3 (e.g., M202 cells) and added the molecular size makers on all the panels in the revised manuscript. We also removed the original Fig. 3D and replaced it with western blot analysis of LNK co-immunoprecipitation and GST-pull down.

2. #1. PD-1 expression and changes in PD-1 levels with IFN γ treatment +/- LNK deletion, should be shown for the B16/F10 cell line used in Fig. 6D. This could be done in cell culture to complement the in vivo experiments, although it would be better if expression analysis was performed on excised tumours.

We assume that the reviewer is referring to the murine Pd-I1 (CD274) instead of Pd-1 (Pd-1 is mainly expressed on T cells and Pd-I1 is expressed on tumor cells). According to the reviewer's suggestion, we performed western blot analysis of the B16/F10 cells treated with recombinant mouse interferon γ . The Pd-I1 protein expression

(InVivoMAb anti-mouse PD-L1 (B7-H1), Bio X cell) in Lnk silenced cells was similar to the control cells upon mouse IFN- γ treatment (30 min or 24 hours, Supplementary Figure 10C).

As suggested by the reviewer, we also tried to perform western blot analysis using total protein lysate extracted from the xenograft tumor tissues. The result was similar to the western blot of the cell line in vitro (Unfortunately, very high background occurred and the bands were masked by the dark background).

2. #2. Fig. 6D is potentially an important result and although there is a trend towards reduced tumor growth with PD-1 antibody and LNK deletion, it is not significant. I appreciate that the authors may have chosen B16F10 cells as they are only moderately responsive to PD-1 blockade, however it may be necessary to try different cell numbers or a different cell line. Ideally these experiments would have been performed using a BRAF mutant B16 cell line which should have elevated levels of LNK and make the effects of LNK deletion more robust (this would also support the first part of the manuscript).

As suggested, we repeated the B16/F10 xenograft experiment with reduced number of cells (0.05 million). The results consistently showed decrease tumor size in the Lnk silenced tumors (in both Pd-1 antibody treatment and non-treatment groups). However, the difference between silencing of Lnk vs control group still did not reach statistical significance (Supplementary Figure 10A). We obtained a murine melanoma cell line harboring a Braf mutation (D4M.3A¹). This cell line was established four years ago from a conditional mouse model of metastatic melanoma: *Tyr::CreER;Braf^{CA};Pten^{lox/lox}*. We generated stable knockdown of Lnk in the D4M.3A cells using lentiviral CRISPR-Cas9 and repeated the Pd-1 experiment in immunocompetent C57BL mice. The results indicated that silencing Lnk in these cells significantly reduced the tumor size and enhanced the effect of Pd-1 treatment. The results have been added to the revised manuscript (Figure 6 D).

2. #3. It is important to know how efficient the LNK deletion is in the B16 cells.

During the revision, we tried many different antibodies against murine Lnk protein and eventually found that the (A-12) Lnk monoclonal antibody (sc-393709) developed recently by Santa Cruz Biotechnology has the best performance and can efficiently detect the endogenous murine Lnk protein (1:250 dilution) (Figure 2D).

We believe the Lnk loci have been extensively silenced in both B16/F10 and D4M.3A cells: #1, we have performed PCR reaction to detect any potential indel caused by the CRISPR mediated knockdown. No PCR amplification band from DNA of Lnk knockout cells could be detected while a clear band was observed in DNA extracted from control cells. #2, Western blot analysis with Lnk antibody using high sensitivity HRP substrate (SuperSignal West Femto Maximum Sensitivity Substrate) and long exposure time (>5 min) consistently show no chemiluminescent signal from the Lnk silenced cells, while a strong band was observed in the control cells.

2. #4. In addition to tumor weight at the experimental endpoint, the authors could measure tumor size over time.

According to the reviewer's suggestion, we measured the tumor sizes over time in the experimental and control mice in new experiments and the results have been added to the revised manuscript (Supplementary Figure 10).

Other comments:

3.1 It is unclear why is there more pSTAT1 in control cells at 24 hours vs 30 min (Fig. 3B)? Usually, Stat phosphorylation peaks within 1 h.

We completely agree with the reviewer that it is unusual that the p-STAT1 signal lasts for 24 hours in M229 cells. Nevertheless, we repeated the experiment during the revision and still observed similar results. Perhaps the continuous activation of p-STAT1 at 24 hours is due to the secondary effect of other cytokines and chemokines induced by the IFN signaling.

Activated IFNG signaling can induce expression of many proinflammatory cytokines and chemokines (can be seen in our RNA sequencing data)^{2, 3}, which may act through an autocrine manner and activated the JAK-STAT1 pathway^{4, 5} even at 24 hours.

3.2 Also, there appears to be no reduction in LNK levels with sgLNK at this timepoint in serum starved cells?

We repeated the experiment and continue to observe LNK at 24 hours in LNK silenced M229 cells grown in serum starved conditions. We suggest the following: Silencing LNK leads to elevated IFN signaling and the addition of IFN signaling caused further stimulation of the expression of *LNK* mRNA (through IRF1, Figure 6) at 24 hours after the treatment, especially in serum free media (see also the answer of question 5). These observations suggest that as a key adaptor protein regulating the strength of cytokines receptor signaling, LNK expression was tightly and dynamically controlled in cells.

4. The authors should show the level of LNK reduction with overexpression and guide deletion in MC38 and B16/F10 cell lines used in experiment Fig. 2C (also relevant for Fig. 6D)

As suggested, we have performed western blot analysis to confirm the overexpression or silencing of LNK before the xenograft experiments using M368, M229 and B16/F10 cells (Figure 2C and Figure 6D).

5. The authors should comment on why they have performed experiments with and without FBS (Fig. 3).

We previously found that LNK protein levels were significantly increased during serum starvation⁶. To preclude the possibility that fetal bovine serum (FBS) (which contain a lot of different cytokines/growth factors) may affect the expression of LNK and its downstream signaling, we performed the experiment with or without FBS in a few cell lines. The result indicates that the effect of LNK in regulating IFN-JAK-STAT is highly consistent in both

culture conditions. We have include the above explanation in Figure 3 legend in our revised manuscript.

6. The authors should check the labeling of "ctrl" vs "GFP in Fig. 3E.'

Thank you. In this experiment we used cell overexpressing GFP as control. We have revised the text and figure legend to be clear.

7. Typographical error in axis labels "Eexpression" in Fig. 4B.

Thank you, we have corrected the typos in the revised manuscript.

Reviewer #2, Expertise: LNK, JAK, STAT (Remarks to the Author):

In this manuscript Ding et al presented a very interesting observation that the expression of LNK is elevated in melanoma that is correlated to hyperactive RAS/RAF/MEK signaling. Overexpression of LNK inhibits IFN/STAT1 signaling and reduces the tumor killing effects of IFN. Reversely, silencing LNK expression enhances IFN signaling and potentiates IFN killing effects. Furthermore, they presented evidence showing that LNK expression is upregulated by IFN signaling as a negative feedback mechanism. This work extends the knowledge of LNK inhibition of JAK/STAT signaling in hematopietc cells, and uncovers a novel role for LNK in melanoma that can be explored for melanoma immunotherapy.

The manuscript performed a comprehensive study of LNK in melanoma. It utilized various public databases to their advantages, multiple cell lines, and more importantly, xenotransplantation of melanoma cell lines manipulated of LNK levels. Although this work is very interesting and potentially of high impact, it is concerning that some statistics are borderline significant or not significant. Furthermore, it heavily relies on overexpression (OE). Knockdown or knockout of LNK have less of a pronounced effect than OE.

1. Figure 1E is a crucial figure for this work, but it used one antibody and no quantification was presented. The authors should validate this antibody

with KD/KO. How to normalize and compare IHC intensities from different batches of tissues should be discussed.

We had performed LNK antibody validation using LNK KD cells before the IHC experiments ⁶ (see below, attached pictures). We tested two LNK antibodies (Santa Cruz (C20) sc-19743 and R&D, AF5888) with different antibody dilutions using LNK silenced cells. The R&D LNK antibody performed better and 1:500 dilution is the best condition for the IHC staining after fixing with 10% formalin overnight followed with 70% ethanol. In melanoma tissue arrays (ME242a, obtained from Biomax Inc), all of the melanoma section cores (n=12) were heavily stained (score =3), while considerable less staining was observed in normal skin tissue (score =1-2). For the IHC staining experiment of 163 melanoma cases, we used the tumor tissue microarray established at the Melanoma Institute Australia. All the tumor samples (n=163) were uniformly processed and immobilized in 4 microarray chips. The tissue microarrays were stained at the same time with the same staining condition. The array chips were independently scored by two pathologists at the Melanoma Institute Australia. Strong staining of LNK was observed (score= 3 or 2) in the majority of the melanoma samples (132 samples, 81%), suggesting LNK protein was indeed highly expressed in most of the melanoma samples. In agreement with our observation, strong IHC staining of LNK protein in melanoma tissues were reported in The Human Protein Atlas database (<https://www.proteinatlas.org/ENSG00000111252-SH2B3/pathology/tissue/skin+cancer#ihc>) using another LNK antibody (Sigma HPA005483).

Non-target shRNA control

shLNK

Figure Legend. IHC staining result of LNK silenced (shLNK) cells using R&D AF5888 LNK antibody.

2. The correlation between LNK expression levels and RAS/RAF mutation status is interesting, but it is correlative. It is not sufficient to conclude that the RAS/RAF/MEK pathway upregulates LNK expression. As controls, the authors should apply other inhibitors to see if LNK expression is specifically linked to RAS pathway. More importantly, it will be compelling to examine if inhibitors that can reduce cell proliferation, could reduce LNK expression. In another word, LNK expression is correlated with cell proliferation or survival, rather than the RAS/RAF pathway?

Thank you for your suggestion. We treated the cells with cisplatin, a chemotherapy drug widely used in cancer treatment. Cells treated with cisplatin caused DNA crosslinking/DNA damage and triggered cell death. Based on the literature and cancer cell lines drug sensitivity data from The Genomic of Drug sensitivity in Cancer database (<https://www.cancerrxgene.org/translation/CellLine/906793>), melanoma cell line A375 (BRAF V600E) is extremely sensitivity to BRAF inhibitor (IC50 z score = -3.8) and cisplatin (IC50 z score = -2.1, 1.36 μ M). We therefore treated the A375 cells with 1 μ M cisplatin for 24 hours, and the LNK protein expression was examined using western blot analysis. The drug significantly reduced proliferation of A375 cells. At the same time, LNK expression was also slightly decreased. This observation suggest that, as pointed out by the reviewer, the cell survival/proliferation may also correlated with LNK expression. We are still investigating the detail mechanism of this

observation; and therefore, we changed in the text from “attributed” to “correlated” (“LNK expression was correlated with hyperactivated RAS/RAF signaling”) and replaced the solid line to a dashed line in the diagram (Figure 6I) in our revised manuscript.

3. #1 Figure 2D showed a bit variable results on the effects of LNK OE or KO. Were the tumors that grew similarly to control has less LNK knockdown or knockout? The authors should show the KO/KD efficiency.

The stable Lnk KD cell line had been generated by lentiviral transduction. We performed western blot analysis to confirm the knock-down efficiency of LNK before the xenograft experiment (Figure 2D in the revised manuscript).

Growth of tumor cells in immunocompetent mice often have bigger variation of tumor size compared to the same cells grown in immunodeficient mice¹. As discussed in the answer of the first reviewer’s question 2.3, we believe the Lnk loci have been extensively silenced in both B16/F10 and D4M.3A cells. We also tried western blot analysis of LNK in the xenograft tumors, unfortunately, the protein quality was not good and we were unsuccessful with western blot (very high dark background).

3. #2 What does Figure 2E “Anoikis” measure? Figure 2F showed higher% of PI+AnnexinV- cells in the GFP control, that is not consistent with apoptosis. PI+AnnexinV- are simply dead cells or debris, and not accurate for quantification.

Anoikis refers to cell death occurring in anchorage-free cells when they are detached from the surrounding extracellular matrix⁷. The Figure 2 E “Anoikis” measured the percentage of alive cells (PI- and annexin V-) after they were grown in anchorage-free conditions (ultralow binding plate). We agree with the reviewer that the pattern of cell death in our Figure 2F was not consistent with the pattern of apoptosis. Nevertheless, staining the cells using another cell viability dye (trypan blue exclusion) consistently suggests that force expression of LNK enhances the survival of cells grown in the anchorage-free conditions. We have revised the text accordingly and replaced the word “apoptosis” to “cell death” in our revised manuscript.

4. Western blots in Figure 3 seem to be a bit variable in different cell lines. pSTAT1 and PD-L1 level changes in A375 seem to be inconsistent with the authors hypothesis, or different from other cell lines.

We agree with the reviewer that the effect of IFNA, IFN beta-1A (IFN-B1A) and beta-1B (IFN-B1B) is slightly inconsistent. This may be due to the following reasons: #1. As detailed in answer to questions Q6 and Q7, we noticed during the revision that the IFN concentration should be reduced in the presence of LNK silenced cells (see answer of questions 6 and 7). #2. This phenomenon may be caused by an indirect effect of other cytokines: IFN treatment often induces expression of many pro-inflammatory cytokines/chemokines^{2, 3}, and the abnormal p-STAT1 and PD-L1 expression at 24 hours may be caused by the secondary effect of those cytokines/chemokines. To avoid confusion, we removed the Figure from our revised manuscript.

5. Figure 3F, Co-IP experiments showed that LNK interacts with JAK1, pSTAT1, and PTPN2. Are they indirect or direct interactions? Can LNK bind to all of them at the same time as one complex?

The direct protein interaction between JAK1 and STAT1, STAT1 and PTPN2, as well as JAK1 and PTPN2 have been shown previously^{8, 9}; thus, we focused our study on the interaction of LNK with STAT1, JAK1 and PTPN2. Reciprocal immunoprecipitation experiment using STAT1 antibody successfully pulled down the endogenous LNK protein together with endogenous STAT1 (Figure 3) and JAK1 protein. Unfortunately, the reciprocal immunoprecipitation experiment using JAK1 and PTPN2 antibodies were unsuccessful. For example, we tried three JAK1 antibodies and none of them were able to pull down the endogenous JAK1 protein. Due to the time limitation of the revision, we removed the PTPN2 and JAK1 results from our manuscript and only focused on LNK's interaction with STAT1. We are currently performing SILAC (Stable isotope labeling by amino acids) Mass Spectrometry to identify comprehensively LNK binding partners in melanoma cells.

To prove direct interaction between LNK and STAT1, we performed GST pull down using purified GST fusion protein. In this study, we used TXK1,

an *E.coli* strain (Stratagene) carrying a plasmid that encodes an inducible tyrosine kinase gene to generate a recombinant protein. To prove that LNK directly binds STAT1, we first expressed and purified recombinant GST fused to the LNK PH domain (GST-PH-LNK) or LNK SH2 domain (GST-SH2-LNK). Recombinant proteins were then incubated with whole cell lysate of HEK293T cells transfected with a Flag tag STAT1. The GST fusion protein and its binding partners were pulled down using glutathione sepharose 4B beads. Both PH and SH2 domain of LNK GST fusion protein, but not the GST control, pulled down the STAT1 protein. In addition, we expressed and purified a recombinant GST-STAT1 (pGEX-STAT1) protein and performed the reciprocal pull down experiment. The results consistently confirmed that only the pGEX-STAT1, but not the control GST protein, pulled down the LNK protein expressed in HEK293T cells. Taken together, the data suggest LNK directly binds to STAT1 protein.

6. It would be great to include RNAseq data with sgRNA to LNK, in addition to OE.

According to the reviewer's suggestion, we performed RNA sequencing using LNK silenced cells (A375) treated either with or without IFN gamma (400 U/ml) for 24 hours. We used two LNK CRISPR-Cas9 silenced cells (sgLNK-1 and sgLNK-2, Supplementary Figures 7A,B). Results showed a remarkable enrichment of the expression signature of interferon genes [#1 enriched pathway (Supplementary Figure 7C), with a false discovery rate (FDR) $q=0$ in Gene Set Enrichment Analysis (GSEA) result]. Other prominent signatures included allograft rejection and immune effectors. We have included these results in our revised manuscript (Figure 5, Supplementary Figures 7, 8).

7. Figure 5 A, it looks that there are some differences between two different sgRNAs to LNK. sgRNA1 to LNK was not significant in some genes (eg, CD274). The genes in the first row showed a higher percentage change in OE than LNK KO, whereas the second row showed a more pronounced difference in LNK KO than LNK OE. Can authors explain this?

During the revision of the manuscript, we repeated the experiments and tested the cellular response using different concentrations of interferon gamma. When the concentration of IFN was reduced from 2000 U/ml to

400 U/ml, the silencing effect of LNK became much more compelling producing a bigger difference between LNK silenced cells vs controls cells. Therefore, we performed the RNA sequencing using cells treated with reduced concentration of IFN gamma; and the results were highly consistent using CRISPR-Cas9 sgLNK1 and sgLNK2 (Figure 5, Supplementary Figure 8).

In addition, we hypothesize that the IFN-JAK/STAT1 signaling pathway is tightly regulated by a number of feedback pathways, and correct timing of harvesting of the cells is critical. We removed the real time PCR result of CD274 in our revised manuscript and replaced it with the RNA sequencing data (Figure 5C).

8. Figure 6A lacks statistical analysis.

As suggested, we have performed and added the statistical analysis (Two-way ANOVA) in the revised Figure 6A.

9. Figure 6D is an important piece of data, but it looks that with PD-1 antibody, there is no significant difference between control and Lnk knockout?

We repeated the B16/F10 experiment with reduced tumor cell number. Results consistently showed retarded tumor growth in Lnk silencing cells (\pm Pd-1 antibody therapy), but the difference did not reach statistical significant (Supplementary Figures 10 A, B). However, we repeated the experiments using a different mouse melanoma cells line model (D4M.3A, carrying a Braf activation mutation). Silencing of Lnk showed a significantly reduced tumor burden, and an almost complete rejection of tumor growth was observed in Lnk silenced cells treated with Pd-1 antibody (Figure 6).

10. The authors showed that IRF-1 may directly bind to LNK promoter and regulate LNK expression. How to explain their data showing that RAS/MAPK pathway upregulates LNK expression?

We hypothesized that in normal condition, BRAF/RAS/MAPK signaling was correlated with the relatively high basal level of LNK expression in

melanoma cells (Figure 1, compared with other type of cancers). Upon exogenous IFN stimulation, LNK mRNA transcript was further increased by IRF1 to suppress the JAK/STAT signaling pathway. This negative feedback prevents the over-activation of the IFN-JAK/STAT pathway. LNK act as a key effector, fine tuning the magnitude and duration of IFN and cytokines.

Reviewer #3, Expertise: melanoma, immune therapy (Remarks to the Author):

Dr. Ding and colleagues suggest a role for LNK in mediating IFN γ signaling and suppressing antitumor immunity. This is a nice contribution although I have a few comments.

Major comments:

1. Several groups have found that MHC-II expression correlates with response to anti-PD-1 (Johnson et al Nat Comm 2016) and tumor intrinsic cell expression of HLA-DR (in similar CCLE analysis) correlated with pro-immune gene expression signatures, even in the absence of infiltrating immune cells. Does LNK expression (in CCLE or in tumors) correlate with MHC-II (HLA-DR) or MHC-I (e.g. HLA-A)?

Thank you for the suggestions. We read the recommend papers and have included the suggested points and cited these references in the Discussion section. Examination for a potential correlation of expression of LNK and MHC molecules, using gene expression data of TCGA melanoma and RNA sequencing data of pre-treatment melanoma samples (GSE78220). We found LNK mRNA expression was indeed negatively correlated with MHC-I and MHC-II molecules in melanoma samples in TCGA and GSE78220 patient cohort (although the correlation is generally weak, with a Pearson correlation value from -0.1 to -0.3, Supplementary Figure 11). This observation is in agreement with our observation that LNK regulates the expression level of MHC-I and MHC-II in melanoma cells treated with IFN γ .

2. Are there mutations in LNK in TCGA melanoma samples and if so, are they gain of function/loss of function? Do mutated samples have PDL1, CD8, MHC-I/II expression?

In melanoma sequencing project done by the Broad Institute (Cell, 2012¹⁰), no mutation was detected in the LNK gene locus in 121 melanoma samples [data retrieved from cBioPortal for Cancer Genomics (www.cbioportal.org)]. In the TCGA melanoma cohort (471 patient/479 samples), only three missense mutations of LNK (A5V, TCGA-D3-A51G; R508P, TCGA-EE-A3AB; T327M, TCGA-ER-A19E) were found. Considering the extremely low mutation rate of LNK [in contrast to the high mutational background of melanoma (highest among all the cancer¹¹)] and the non-recurrent mutational pattern, the three non-recurrent missense mutations are highly likely passenger mutations. This observation is in agreement with our proposed function of LNK in melanoma: LNK act as a pro-tumor gene with high expression in melanoma. PDL1 and CD8 are only barely expressed in these three LNK mutated melanoma samples, while MHC-I and MHC-II are modestly expressed. The expression level of these genes in these three samples were not different than the rest melanoma samples.

3. #1 Figure 6D - was there any difference in survival in these mice? Any difference in T cell infiltration? At what timepoint were these tumors harvested?

We injected control and Lnk silenced cells in the same mouse (control cells on one flank and Lnk silenced cells on the other flank) and harvested the tumors (both control and experimental groups) at the same time (day 18 for B16/F10 and day 23 for D4M.3A melanoma cells). A schematic diagram was added in the revised manuscript to indicate the Pd-1 injection day and tumor harvest day (Figure 6E). Unfortunately, our Institutional Animal Care and Use Committee (IACUC) required us to euthanize mice once the tumor size reach 1.5 cm; thus, we are not allowed directly to compare survival of the mice. Nevertheless, since the tumor size are smaller in the Lnk silenced tumors, we believe the mice carrying the Lnk silenced tumors will survive longer if we were allowed to sacrifice the mice separately when their tumors reached 1.5 cm.

According to the reviewer's suggestion, we examined the T cell marker CD3 by RT-PCR using RNA from the tumor tissues of the controls and Lnk silenced cells. A 2 fold increased expression of CD3 occurred in the Lnk silenced tumors (Supplementary Figure 10D), suggesting an increased T

cell infiltration in tumors formed from Lnk knockdown cells. No further increase of CD3 expression occurred in tumors from mice treated with Pd-1 antibody, supporting the concept that Pd-1 therapy mainly reinvigorates the exhausted T cells instead of increasing the T cell infiltration into the tumors.

3. #2 Is this a known anti-PD-1 sensitive or resistant model? Just showing differences in tumor size is not very convincing.

B16/F10 is known as a poor immunogenic cell line¹² and it poorly responds to PD-1 therapy¹³. PD-1 blockade alone is ineffective in C57BL/6 mice harboring the B16/F10 tumors¹⁴, unless the mice were immune stimulated with a cancer vaccine such as GVAX¹⁵.

Minor comments:

1. Introduction: the rate of durable responses to anti-PD-1 is 30-40% not 20-30%.

Thank you for this point, we accordingly revised the text.

2. Similarly, the rate of secondary resistance is closer to 20-33% rather than 40-50%

Again, thank you. We revised the text.

3. Figures 3C-E - Please make it more clear which panels have forced expression of LNK and which have LNK silenced.

We apologize for the confusion. We have revised the Figures and added a note below each western blot to indicate either LNK overexpression or LNK knockdown status.

4. Page 11 - should note that adjuvant melanoma treatment has been interferon alpha not interferon gamma

Thank you for pointing this out. We have added the references and accordingly placed a note on the corresponding text.

References

1. Jenkins MH, *et al.* Multiple murine BRAf(V600E) melanoma cell lines with sensitivity to PLX4032. *Pigment Cell Melanoma Res* **27**, 495-501 (2014).
2. Faggioli L, *et al.* Molecular mechanisms regulating induction of interleukin-6 gene transcription by interferon-gamma. *Eur J Immunol* **27**, 3022-3030 (1997).
3. Harroch S, Revel M, Chebath J. Induction by interleukin-6 of interferon regulatory factor 1 (IRF-1) gene expression through the palindromic interferon response element pIRE and cell type-dependent control of IRF-1 binding to DNA. *EMBO J* **13**, 1942-1949 (1994).
4. Wang L, *et al.* IL6 Signaling in Peripheral Blood T Cells Predicts Clinical Outcome in Breast Cancer. *Cancer Res* **77**, 1119-1126 (2017).
5. Canaff L, Zhou X, Hendy GN. The proinflammatory cytokine, interleukin-6, up-regulates calcium-sensing receptor gene transcription via Stat1/3 and Sp1/3. *J Biol Chem* **283**, 13586-13600 (2008).
6. Ding LW, *et al.* LNK (SH2B3): paradoxical effects in ovarian cancer. *Oncogene* **34**, 1463-1474 (2015).
7. Gilmore AP. Anoikis. *Cell Death And Differentiation* **12**, 1473 (2005).
8. ten Hoeve J, *et al.* Identification of a nuclear Stat1 protein tyrosine phosphatase. *Mol Cell Biol* **22**, 5662-5668 (2002).
9. Kleppe M, *et al.* PTPN2 negatively regulates oncogenic JAK1 in T-cell acute lymphoblastic leukemia. *Blood* **117**, 7090-7098 (2011).
10. Hodis E, *et al.* A landscape of driver mutations in melanoma. *Cell* **150**, 251-263 (2012).
11. Martincorena I, Campbell PJ. Somatic mutation in cancer and normal cells. *Science* **349**, 1483-1489 (2015).

12. Celik C, Lewis DA, Goldrosen MH. Demonstration of immunogenicity with the poorly immunogenic B16 melanoma. *Cancer Res* **43**, 3507-3510 (1983).
13. Kreiter S, et al. Mutant MHC class II epitopes drive therapeutic immune responses to cancer. *Nature* **520**, 692-696 (2015).
14. Juneja VR, et al. PD-L1 on tumor cells is sufficient for immune evasion in immunogenic tumors and inhibits CD8 T cell cytotoxicity. *J Exp Med* **214**, 895-904 (2017).
15. Curran MA, Montalvo W, Yagita H, Allison JP. PD-1 and CTLA-4 combination blockade expands infiltrating T cells and reduces regulatory T and myeloid cells within B16 melanoma tumors. *Proc Natl Acad Sci U S A* **107**, 4275-4280 (2010).

REVIEWERS' COMMENTS:

Reviewer #1 (Remarks to the Author):

The authors have made a significant effort to address the reviewer's comments and this has resulted in a much improved manuscript.

RE: the addition of MW markers. The authors should actually place a mark where the MW std was located, rather than just put a number next to the western blot.

Reviewer #2 (Remarks to the Author):

The manuscript has markedly improved in the revised form. This is a novel report that is potentially important for immune therapy. Although many questions remain unanswered regarding how LNK modulates IFN responses in melanoma development, this report identified a potentially important direct and mechanism in the field.

Wei Tong

Reviewer #3 (Remarks to the Author):

My concerns have been addressed

REVIEWERS' COMMENTS:

Reviewer #1 (Remarks to the Author):

The authors have made a significant effort to address the reviewer's comments and this has resulted in a much improved manuscript.

RE: the addition of MW markers. The authors should actually place a mark where the MW std was located, rather than just put a number next to the western blot.

According to the suggestion, we placed line marks to indicate precisely the molecular weight in our revised manuscript.

Reviewer #2 (Remarks to the Author):

The manuscript has markedly improved in the revised form. This is a novel report that is potentially important for immune therapy. Although many questions remain unanswered regarding how LNK modulates IFN responses in melanoma development, this report identified a potentially important direct and mechanism in the field.

Wei Tong

Reviewer #3 (Remarks to the Author):

My concerns have been addressed